# Independence Tests for Language Models

**Sally Zhu** [* 1]  **Ahmed Ahmed** [* 1]  **Rohith Kuditipudi** [* 1]  **Percy Liang** [1]

## Abstract

Motivated by liability and intellectual property concerns over open-weight models we consider the following problem: given the weights of two models, can we test whether they were trained independently—i.e., from independent random initializations? We consider two settings: *constrained* and *unconstrained*. In the constrained setting, we make assumptions about model architecture and training and propose statistical tests that yield exact p-values with respect to the null hypothesis that the models are trained from independent random initializations. We compute the p-values by simulating exchangeable copies of each model under our assumptions and comparing various similarity measures between the original two models versus these copies. We report p-values on pairs of 21 open-weight models (210 total pairs) and find we correctly identify all pairs of non-independent models. In the unconstrained setting we make none of the prior assumptions and allow for adversarial evasion attacks that do not change model output. We thus propose a new test which matches hidden activations between two models, which is robust to these transformations and to changes in model architecture and can also identify specific non-independent components of models. Though we no longer obtain exact p-values from this test, empirically we find it reliably distinguishes non-independent models like a p-value. Notably, we can use the test to identify specific parts of one model that are derived from another (e.g., how Llama 3.1-8B was pruned to initialize Llama 3.2-3B, or shared layers between Mistral-7B and StripedHyena-7B), and it is even robust to retraining individual layers of either model from scratch.

*Equal contribution [1]Department of Computer Science, Stanford University, Stanford, US. Correspondence to: Sally Zhu <salzhu@stanford.edu>, Ahmed Ahmed <ahmedah@cs.stanford.edu>.

*Proceedings of the $42^{nd}$ International Conference on Machine Learning*, Vancouver, Canada. PMLR 267, 2025. Copyright 2025 by the author(s).

## 1. Introduction

Consider the ways in which two models could be related: one model may be a finetune of the other; one could be spliced and pruned from certain parts of the other; both models could be separately fine-tuned from a common ancestor; finally, they could be independently trained from each other. We consider the problem of determining whether two models are independently trained versus not from their weights, which we formalize as a hypothesis testing problem in which the null hypothesis is that the weights of the two models are independent. We concretely treat only the weight initialization as random and thus consider two models with different random initial seeds as independent, even if both models were trained on the same data, or one model was distilled from the outputs of the other.

A solution to this independence testing problem would help auditors track provenance of open-weight models. This is pertinent because while open-weight models enable broader access and customization, they also pose potential risks for misuse as they cannot be easily monitored or moderated (Kapoor et al., 2025). Model developers would also gain an enhanced ability to protect their intellectual property (IP) (Mensch, 2024; Peng et al., 2023) and enforce custom model licenses (Dubey et al., 2024; DeepSeek-AI et al., 2024).

We consider two settings of the independence testing problem. In the constrained setting, we make assumptions on training and initialization (essentially, that the training algorithm is equivariant to permuting the hidden units of the random initialization) that enable us to obtain provably valid p-values. The main idea is that under these assumptions we can cheaply simulate many exchangeable copies of each model's weights and compare the value of some test statistic (e.g., cosine similarity of model weights) on each of these copies with the original model pair. The assumptions generally hold in practice but preclude robustness to adversarial evasion attacks and architectural changes.

For the constrained setting, we evaluate various test statistics on 21 models of the Llama 2 architecture (Touvron et al., 2023), including 12 fine-tunes of Llama 2 and nine independently trained models, obtaining extremely small p-values

We share code at https://github.com/ahmeda14960/model-tracing.

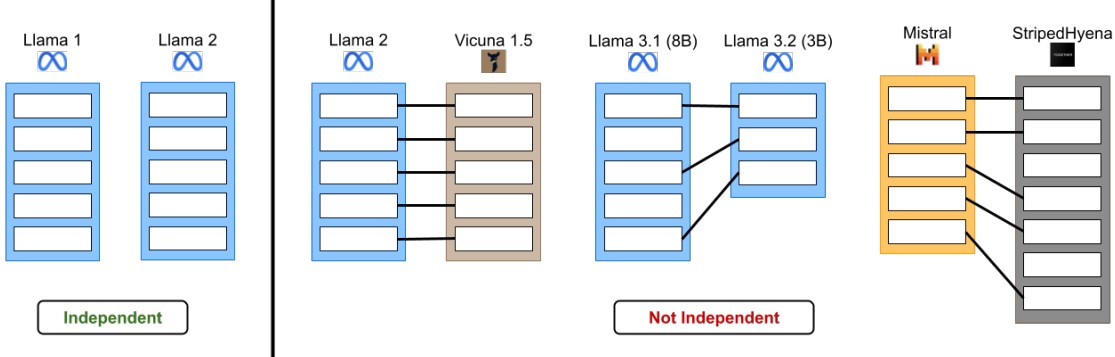

*Figure 1.* Given the weights of two models, what relationships can we derive?

for all 69 non-independent model pairs. Notably, our tests retain low p-values over different fine-tuning methods (e.g., different optimizers) and on models fine-tuned for many tokens from the base model such as Llemma (Azerbayev et al., 2024), which was fine-tuned on an additional 750B tokens from Llama 2 (i.e., 37.5% of the Llama 2 training budget). We also confirm that the leaked Miqu-70B model from Mistral is derived from Llama 2-70B.

For the unconstrained setting we develop a test robust to simple modifications to model weights and architecture, such as permuting hidden units, that can violate the assumptions of the constrained setting if an adversary applies them after fine-tuning. Though we are not able to obtain provably exact p-values in the unconstrained setting, we derive a test whose output empirically behaves like a p-value and reliably distinguishes non-independent models from independent models. In particular, we first align the hidden units of two models—which may each have different activation types and hidden dimensions—and then compute some measure of similarity between the aligned models. Because of the alignment step, the test is robust to changes in model architecture and various adversarial evasion attacks (including those that break prior work). Moreover, it can localize the dependence: we can identify specific components or weights that are not independent between two models, even when they have different architectures.

We evaluate our unconstrained setting test on 141 independent model pairs and find that its output empirically behaves like a p-value in the sense that it is close to uniformly distributed in $[0, 1]$ over these pairs. In contrast, it is almost zero for all dependent pairs we test (including those for which we simulate a somewhat strong adversary by retraining entire layers from scratch). We also employ our test to identify *pruned* model pairs, which occur when one reduces the layer dimensions by retaining only select activations and weights from a pre-trained model; for example, we identified the precise layers of Llama 3.1 8B from which each Llama 3.2 3B and Llama 3.2 1B layer was derived.

The work most closely related to ours is due to Zeng et al. (2024), who considered our constrained setting; they develop various tests to determine whether a model as a whole is independent of another by computing the cosine similarity of the products of certain weight matrices in both models. They show that their tests are robust to simple adversarial transformations of model weights that preserve model output; however, we detail in Appendix G.1 other transformations to perturb dependent models that evades detection by their tests. Additionally, unlike Zeng et al. (2024), in the constrained setting we obtain exact p-values from our tests. Jin et al. (2024) propose crafting specific queries that are likely to produce different responses among independently trained models; their method does not require access to weights but also does not produce exact p-values.

## 2. Methods

### 2.1. Problem formulation

Let $f : \Theta \times \mathcal{X} \to \mathcal{Y}$ denote a *model* mapping parameters $\theta \in \Theta$ and an input $X \in \mathcal{X}$ to an output $f(X; \theta) \in \mathcal{Y}$. We represent a model training or fine-tuning process as a *learning algorithm* $A : \Theta \to \Theta$ that takes as input a set of initial parameters corresponding to either a random initialization or, in the case of fine-tuning, base model parameters. Specifically, $A$ includes the choice of training data, ordering of minibatches, and all other design decisions and even the randomness used during training—everything other than the initial model weights.

Given two models $\theta_1, \theta_2 \sim P$ for some joint distribution $P \in \mathcal{P}(\Theta_1 \times \Theta_2)$, our goal is to test the null hypothesis

$$H_0 : \theta_1 \perp \theta_2, \tag{1}$$

where $\perp$ denotes independence of two random variables. One example of a case where $\theta_1$ and $\theta_2$ might not be independent is if $\theta_2$ is fine-tuned from $\theta_1$, i.e., $\Theta_1 = \Theta_2$ (meaning the two models share the same architecture) and $\theta_2 = A(\theta_1)$ for some learning algorithm $A$. We treat

learning algorithms as deterministic functions. Thus, for $\theta_1 = A_1(\theta_1^0)$ and $\theta_2 = A_2(\theta_2^0)$, then $\theta_1^0 \perp \theta_2^0$ (i.e. two models with independent random initializations) implies our null hypothesis.

Deep learning models are often nested in nature. For example, Transformer models include self-attention layers and MLP layers as submodels. We formalize the notion of a submodel via the following definition.

**Definition 1.** A model $f : \Theta \times \mathcal{X} \to \mathcal{Y}$ *contains* a submodel $g : \Theta' \times \mathcal{X}' \to \mathcal{Y}'$ if there exists a projection operator $\text{proj} : \Theta \to \Theta'$ such that for all $\theta \in \Theta$ we have

$$f(x; \theta) = f_{\text{out}}(g(f_{\text{in}}(x); \text{proj}(\theta)))$$

for some functions $f_{\text{in}} : \mathcal{X} \to \mathcal{X}'$ and $f_{\text{out}} : \mathcal{Y}' \to \mathcal{Y}$ (which may depend on $\theta$).

Many of our experiments will involve Transformer models specifically containing MLP layers with Gated Linear Unit (GLU) activations, which are widely used among language models. It thus will be useful to define this type of MLP presently through the following example.

**Example 1:** (GLU MLP) Let $G, U \in \mathbb{R}^{h \times d}$ and $D \in \mathbb{R}^{d \times h}$. Let $\sigma : \mathbb{R} \to \mathbb{R}$ be an element-wise activation function. For $x \in \mathbb{R}^d$ and $\theta = (G, U, D) \in \Theta_{\text{mlp}}^h$, let $f_{\text{mlp}}(x; \theta) := D(\sigma(Gx) \odot (Ux))$. Likewise, for $X \in \mathbb{R}^{s \times d}$ let $f_{\text{mlp}}(X; \theta) \in \mathbb{R}^{s \times d}$ denote the result of broadcasting $f_{\text{mlp}}$ over the rows of $X$. $\diamond$

In addition to the basic independence testing problem above, we also consider the problem of *localized testing*: testing whether various pairs of submodels among two overall models are independent or not. A prototypical example of a localized testing problem is identifying which layers of a larger model (e.g., Llama 3.1-8B) were used to initialize a smaller model (e.g., Llama 3.2-3B) (in this case, we treat the layers as different submodels).

### 2.2. Constrained Setting

#### 2.2.1. TESTING FRAMEWORK

Algorithm 1 (PERMTEST) encapsulates our framework for computing p-values against the null hypothesis in the constrained setting, wherein we simulate $T$ exchangeable copies of the first model $\theta_1$ by applying transformations to its weights. The exchangeability of these copies holds under some assumptions on the learning algorithm and random initialization that produced the original model. We capture these assumptions in the following definitions; together, they define the constrained setting.

**Definition 2** ($\Pi$-invariance). Let $\Pi \subset \Theta \to \Theta$. A distribution $P \in \mathcal{P}(\Theta)$ is $\Pi$-*invariant* if for $\theta \sim P$ and any $\pi \in \Pi$, the parameters $\theta$ and $\pi(\theta)$ are identically distributed.

---

**Algorithm 1:** Test for computing p-values (PERMTEST)

**Input:** Model weights $\theta_1, \theta_2$
**Parameters:** test statistic $\phi$; discrete transformation class $\Pi$; permutation count $T$
**Output:** p-value $\hat{p} \in (0, 1]$

1   n_ties $\leftarrow 0$;
2   **for** $t \in 1, \dots, T$ **do**
3      $\pi_t \sim \text{Unif}(\Pi)$;
4      $\phi_t \leftarrow \phi(\pi_t(\theta_1), \theta_2)$;
5      $s \leftarrow s + \mathbf{1}\{\phi_t = \phi(\theta_1, \theta_2)\}$;
6   $\xi \sim \text{Unif}(\{0, \dots, \text{n\_ties}\})$ // break ties
7   $\hat{p} \leftarrow 1 - \frac{1}{T+1}(1 + \xi + \sum_{t=1}^T \mathbf{1}\{\phi_t < \phi(\theta_1, \theta_2)\}$;
8   **return** $\hat{p}$

---

**Definition 3** ($\Pi$-equivariance). Let $\Pi \subset \Theta \to \Theta$, $\pi \in \Pi$, and $\theta^0 \in \Theta$. A learning algorithm $A$ is $\Pi$-*equivariant* if and only if $\pi(A(\theta^0)) = A(\pi(\theta^0))$.

The main idea underlying PERMTEST is that so long as $\theta_1 = A(\theta_1^0)$ and $\theta_1^0 \sim P$ for some $\Pi$-equivariant learning algorithm $A$ and $\Pi$-invariant distribution $P$, we can simulate $T$ exchangeable (but not independent) copies $\{\pi_t(\theta_1)\}_{t=1}^T$ of $\theta_1$ by sampling $\pi_t \overset{\text{i.i.d.}}{\sim} \text{Unif}(\Pi)$. This allows us to efficiently compute an exact p-value without actually repeating the training process of $\theta_1$. In effect, Definitions 2 and 3 imply that $\pi$ commutes with $A$—i.e., $\pi(A(\theta_1^0)) = A(\pi(\theta_1^0))$. Under exchangeability, the p-value output by PERMTEST will be uniformly distributed over $\{(i + 1)/(T + 1)\}_{i=0}^T$.

Standard initialization schemes for feedforward networks are symmetric over the hidden units of the network, and so one example of a class of transformations with respect to which any such initialization is invariant is the set of permutations over the hidden units of the network. Moreover, the gradient of the model's output with respect to the hidden units is permutation equivariant; thus, any learning algorithm whose update rule is itself a permutation equivariant function of gradients (e.g., SGD, Adam, etc.) satisfies Definition 3 with respect to these transformations. A (contrived) example of a learning algorithm that is not permutation equivariant is one that uses different learning rates for each hidden unit depending on the index of the hidden unit.

**Example 2** (Permuting hidden units): Let $\theta = (G, U, D) \in \Theta_{\text{mlp}}^h$ parameterize a GLU MLP, where recall $f_{\text{mlp}}(x; \theta) := D(\sigma(Gx) \odot (Ux))$ for some element-wise activation function $\sigma : \mathbb{R} \to \mathbb{R}$. Abusing notation, let $\Pi$ be the set of $h \times h$ permutation matrices such that for $\pi \in \Pi$ we define $\pi(\theta) = (\pi G, \pi U, D\pi^T)$. Observe $f_{\text{mlp}}(x; \theta) = f_{\text{mlp}}(x; \pi(\theta))$ and $\pi(\nabla_\theta f_{\text{mlp}}(x; \theta)) = \nabla_{\pi(\theta)} f(x; \pi(\theta))$ for all inputs $x$. $\diamond$

The assumptions we make in the constrained setting suffice for PERMTEST to produce a valid $p$-value, as we show in

the following theorem, whose proof uses symmetry of the initialization and training process (full proof in Appendix A).[1] Importantly, the result of the theorem holds (under the null hypothesis) without any assumptions on $\theta_2$; so, a model developer of $\theta_1$ testing other models with our methods can have confidence in the validity of our test without trusting the provider of $\theta_2$. Of course, if $\theta_2$ does not satisfy the equivariance assumption on training (as in the unconstrained setting), then PERMTEST is unlikely to produce a low p-value even in cases where $\theta_1$ and $\theta_2$ are not independent (e.g., if an adversary finetunes $\theta_2$ from $\theta_1$ but then afterwards randomly permutes its hidden units).

**Theorem 1.** *Let* $\phi : \Theta \times \Theta \to \mathbb{R}$ *be a test statistic and* $\Pi \subset \Theta \to \Theta$ *be finite. Let* $A : \Theta \to \Theta$ *be* $\Pi$-*equivariant and let* $P \in \mathcal{P}(\Theta)$ *be* $\Pi$-*invariant. For* $\theta_1^0 \sim P$, *let* $\theta_1 = A(\theta_1^0)$. *Let* $\theta_2 \in \Theta$ *be independent of* $\theta_1$. *Then* $\widehat{p} = \text{PERMTEST}(\theta_1, \theta_2)$ *is uniformly distributed on* $\{\frac{i+1}{T+1}\}_{i=0}^T$.

We also generalize Theorem 1 to apply to randomized learning algorithms that satisfy a notion of equivariance in distribution (including dropout) in Appendix B. However, throughout the main text we will continue to treat learning algorithms as deterministic for the sake of simplicity.

### 2.2.2. TEST STATISTICS

We have shown PERMTEST produces a valid p-value regardless of the test statistic $\phi$ we use. The sole objective then in designing a test statistic is to achieve high statistical power: we would like $\widehat{p} = \text{PERMTEST}(\theta_1, \theta_2)$ to be small when $\theta_1$ and $\theta_2$ are not independent. The statistics in this section apply to any model pair sharing the same architecture.

Prior work (Xu et al., 2024) proposed testing whether two models are independent or not based on the $\ell_2$ distance between their weights, summed over layers. Specifically, for a model with $L$ layers parameterized by $\Theta = \Theta_1 \times ... \times \Theta_L$, with $\theta_1 = (\theta_1^{(\ell)})_{\ell=1}^L$ and $\theta_2 = (\theta_2^{(\ell)})_{\ell=1}^L$, let $\phi_{\ell_2}(\theta_1, \theta_2) := -\sum_{i=1}^L \ell_2(\theta_1^{(\ell)}, \theta_2^{(\ell)})$. We can obtain p-values from $\phi_{\ell_2}$ by using it within PERMTEST. However, a major limitation is that in order to obtain a p-value less than $1/(T+1)$ we must recompute $\phi_{\ell_2}$ at least $T$ times; then the statistical power of our test using $\phi_{\ell_2}$ is therefore bottlenecked by computation.

To address this limitation, we propose a family of test statistics whose distribution under the null is identical for *any* model pair. The test statistics all share the following general form based on Algorithm 2 (MATCH): for $m, n \in \mathbb{N}$ and $M : \Theta \to \mathbb{R}^{n \times m}$, let

$$\phi_M(\theta_1, \theta_2) := \text{SPEARMAN}(\text{MATCH}(M(\theta_1), M(\theta_2)),$$
$$[1, ..., n]), \qquad (2)$$

where SPEARMAN is the Spearman rank correlation (Algorithm 3). Equation (2) is applicable to any model architecture $\Theta$ for which we can define a suitable matrix valued function $M$ of model parameters. For example, $M$ could extract a weight matrix or activation matrix (based on some set of inputs) from a layer of the model, where each row corresponds to a hidden unit of the model. We use MATCH to align the rows of the two extracted matrices and compute the Spearman correlation of this alignment with the identity map between rows. We describe matching in Algorithm 2, wherein cossim denotes cosine similarity function and LAP denotes the algorithm of Ramshaw & Tarjan (2012) we use to solve the matching problem.

[1]As a result of the test yielding exact p-values, we can directly control for the false positive rate via the significance threshold.

The idea is that for two dependent models, each row of $M(\theta_1)$ should be similar to its counterpart in $M(\theta_2)$; thus, the alignment found by SPEARMAN will be close to the identity map. Meanwhile, so long as $M$ is a $\Pi$-equivariant map (Definition 4), then $\phi_M(\theta_1, \theta_2)$ under the null yields valid p-values (see Theorem 2 and proof in Appendix A); so we can use the more computationally-efficient Algorithm 3 to convert statistics to p-values instead of running PERMTEST.

**Definition 4.** (equivariant map) A matrix-valued function $M : \Theta \to \mathbb{R}^{n \times m}$ is $\Pi$-equivariant with respect to a class of transformations $\Pi : \Theta \to \Theta$ if there exists a bijection between $\Pi$ and the set of $n \times n$ permutation matrices such that $M(\pi(\theta)) = \pi M(\theta)$ for all $\theta \in \Theta$ and $\pi \in \Pi$.

**Theorem 2.** *Let* $M : \Theta \to \mathbb{R}^{n \times m}$ *be a* $\Pi$ *equivariant map and let* $P \in \mathcal{P}(\Theta)$ *be* $\Pi$-*invariant. Let* $\theta_1, \theta_2 \in \Theta$ *be independent random variables, with* $\theta_1 = A(\theta_1^0)$ *for* $\theta_1^0 \sim P_1$. *Then* $\phi_M(\theta_1, \theta_2)$ *is uniformly distributed on* $[0, 1)$.

---

**Algorithm 2:** Cosine similarity matching (MATCH)

**Input:** Matrices $W_1, W_2$ with $h$ rows
**Output:** Permutation $\pi : [h] \to [h]$

1 **for** $i \in 1, \ldots, h$ **do**
2      **for** $j \in 1, \ldots, h$ **do**
3          $C_{i,j} \leftarrow \text{cossim}((W_1)_i, (W_2)_j)$;
4 $\pi \leftarrow \text{LAP}(C)$;
5 **return** $\pi$

---

Taking various such functions $M$ yields different test statistics. We focus our experiments on Transformer models consisting of a series of $L$ Transformer blocks that each contain a GLU MLP submodel, and we take $M(\theta)$ to be either the up projection weights or the hidden-layer activations of one of these MLP submodels. In particular, let $U^{(\ell)}(\theta) \in \mathbb{R}^{h \times d}$ denote the first layer up projection weights of the MLP in the $\ell$-th block, where $h$ is the hidden dimension and $d$ is the input dimension, and let $H^{(\ell)}(\theta) \in \mathbb{R}^{h \times (N \cdot s)}$ denote the (flattened) hidden activations that obtain from passing $N$ length $s$ input sequences $X \in \mathbb{R}^{N \times s \times d}$ to the same MLP

module (the test is valid for any $X$; we will specify later how we choose $X$ in our experiments). The two main test statistics we employ in our experiments are $\phi_{U^{(\ell)}}$ and $\phi_{H^{(\ell)}}$.

Both $U^{(\ell)}$ and $H^{(\ell)}$ are equivariant with respect to permuting the hidden units of the corresponding MLP, so we can directly interpret the outputs of $\phi_{U^{(\ell)}}$ and $\phi_{H^{(\ell)}}$ as p-values. Moreover, we can separately permute the hidden units of the MLP in the $\ell$-th block without changing the inputs or outputs of the other blocks. Thus, as we show in Theorem 3 (proof in Appendix A), we can aggregate the p-values from $\phi_{U^{(\ell)}}$ and $\phi_{H^{(\ell)}}$ across blocks using Fisher's method ((Mosteller & Fisher, 1948)) to obtain a more powerful test in Algorithm 4 (`FISHER`).

---

**Algorithm 3:** Deriving p-values from Spearman correlation (`SPEARMAN`)

**Input:** Permutations $\pi_1, \pi_2 : [h] \to [h]$
**Output:** p-value $\hat{p} \in (0, 1]$

1   $r \leftarrow 1 - \frac{6 \sum (\pi_1[i] - \pi_2[i])^2}{h(h^2 - 1)}$;
2   $t \leftarrow r\sqrt{\frac{h-2}{1-r^2}}$;
3   $\hat{p} \leftarrow \mathbb{P}(T_{n-2} > t)$;
4   **return** $\hat{p}$

---

**Algorithm 4:** Aggregating p-values (`FISHER`)

**Input:** p-values $\{\hat{p}^{(i)}\}_{i=1}^{L}$
**Output:** p-value $\hat{p} \in (0, 1]$

1   $\xi \leftarrow \sum_{i=1}^{L} \log \hat{p}^{(i)}$;
2   $\hat{p} \leftarrow 1 - \mathbb{P}(\chi^2_{2L} < -2\xi)$;
3   **return** $\hat{p}$

---

**Theorem 3.** *Consider block indices $i, j \in [L]$ with $i \neq j$ for models with $L$ blocks. Suppose for $\ell \in \{i, j\}$ that*

1. *$M^{(\ell)} : \Theta \to \mathbb{R}^{h \times N}$ is equivariant with respect to $\Pi^{(\ell)}$, i.e., for any $\theta \in \Theta$ and $\pi^{(\ell)} \in \Pi^{(\ell)}$ we have*

$$M(\pi^{(\ell)}(\theta)) = \pi^{(\ell)} M(\theta).$$

2. *$A$ is a $\Pi^{(\ell)}$-equivariant learning algorithm and $P \in \mathcal{P}(\Theta)$ is a $\Pi^{(\ell)}$-invariant distribution.*

*Let $\theta_1, \theta_2 \in \Theta$. If $\theta_1 \perp \theta_2$ for $\theta_1 = A(\theta_1^0)$ with $\theta_1^0 \sim P$, then*

$$\texttt{MATCH}(M^{(i)}(\theta_1), M^{(i)}(\theta_2)) \perp \texttt{MATCH}(M^{(j)}(\theta_1), M^{(j)}(\theta_2)).$$

Recall $\phi_{U^{(\ell)}}$ and $\phi_{H^{(\ell)}}$ are functions of $\texttt{MATCH}(M^{(\ell)}(\theta_1), M^{(\ell)}(\theta_2))$ respectively for $M^{(\ell)} = U^{(\ell)}$ and $M^{(\ell)} = H^{(\ell)}$, both of which satisfy the assumptions of the theorem. Thus, the result of the theorem applies to both

these test statistics, and the independence of the p-values from these test statistics across blocks follows directly from the independence of the statistics themselves.

## 2.3. Unconstrained Setting

For the unconstrained setting, our goal is to design a robust test that applies to models of different architectures and is robust to output-preserving transformations of model weights. Recall our tests for the constrained setting satisfy neither of these desiderata: these tests assume both models have the same number of hidden units, and it is easy to fool them without changing the output of a model by permuting the order of the hidden units in the model.

Our robust test reposes on the design of $\phi_M$ in equation (2). The goal is to identify two matrix valued functions of model parameters $M, M' : \Theta \to \mathbb{R}^{n \times m}$ that jointly satisfy the following condition: any output-preserving transformation of model parameters must transform both $M$ and $M'$ in the same way. Then, whereas previously we would correlate $\texttt{MATCH}(M(\theta_1), M(\theta_2))$ with the identity permutation, we instead define

$$\phi_{M,M'} := \texttt{SPEARMAN}(\texttt{MATCH}(M(\theta_1), M(\theta_2)),$$
$$\texttt{MATCH}(M'(\theta_1), M'(\theta_2)). \quad (3)$$

The above goal is aspirational in the sense that for any nontrivial deep learning model we are not able to fully enumerate the set of transformations of model parameters to which model output is invariant; nonetheless, it will serve as a useful guiding principle for designing our robust test under the framework of equation (3). We organize the description of our full robust test—which is generally applicable to a variety of model architectures—into two parts: first, in Section 2.3.1 we instantiate equation (3) to obtain a test for GLU MLP models. Then, in Section 2.3.2 we use our GLU MLP test as a primitive for designing a test that applies to general deep learning models (including those which do not contain any GLU MLP submodels).

### 2.3.1. TESTING GLU MODELS

Recalling our definition of a GLU MLP model in Example 1, for $k \in \{1, 2\}$ let $\theta_k = (G_k, U_k, D_k) \in \Theta_{\text{mlp}}^{h_k}$, and with inputs $X \in \mathbb{R}^{d \times N}$ let $H_{\text{up}}(\theta_k) = U_k X \in \mathbb{R}^{\max\{h_1, h_2\} \times N}$ be the output of the up projection operation and let $H_{\text{gate}}(\theta_k) = G_k X \in \mathbb{R}^{\max\{h_1, h_2\} \times N}$ be the output of the gate projection operation (with appropriate zero-padding when $h_1 \neq h_2$). Due to the element-wise product operation, we conjecture that in general it is not possible to permute the rows of $G_k$ while preserving the output of $\theta_i$ without permuting the rows $U_k$ in the same way, and so we use $\phi_{M,M'}$ with $M = H_{\text{gate}}$ and $M' = H_{\text{up}}$ for our GLU MLP test. Henceforth, we will shorthand this test as $\phi_{\text{MATCH}}$.

As with the constrained setting, we focus much of our experiments on Transformer models, which recall consist of a series of $L$ Transformer blocks that each contain a GLU MLP submodel. Adopting the notational conventions of Section 2.2.2, we can apply our GLU MLP test to the $\ell$-th block by taking $M = H_{\text{gate}}^{(\ell)}$ and $M' = H_{\text{up}}^{(\ell)}$, where like before (in the case of $\phi_{H^{(\ell)}}$) we obtain the activation inputs for each block by computing a forward pass through the full model over a set of length $s$ sequences of input tokens.

We can aggregate the results of these tests over blocks using FISHER, like we do for $\phi_{U^{(\ell)}}$ and $\phi_{H^{(\ell)}}$ in the constrained setting. Alternatively, we can apply the test to all possible $O(L^2)$ pairs of blocks between two Transformer models if we suspect that certain blocks from one model served as the initializations for different blocks in the other model. Specifically, we can test the $i$-th block of $\theta_1$ and the $j$-th block of $\theta_2$ using

$$\phi_{\text{MATCH}}^{(i,j)} := \text{SPEARMAN}(\text{MATCH}(H_{\text{gate}}^{(i)}(\theta_1), H_{\text{gate}}^{(j)}(\theta_2)),$$
$$\text{MATCH}(H_{\text{up}}^{(i)}(\theta_1), H_{\text{up}}^{(j)}(\theta_2))).$$

This test is relevant for pruned models, where only select blocks (layers) of $\theta_2$ may be used to initialize the smaller $\theta_1$; or, if an adversary takes only certain layers, or even only certain activations, of a pre-trained model and injects other layers.

### 2.3.2. BEYOND GLU MODELS

Thus far we have focused on models $f : \mathcal{X} \times \Theta \to \mathcal{Y}$ containing a GLU MLP submodel. In particular, recalling Definition 1, we have assumed for some $\text{proj}_{\text{mlp}} : \Theta \to \Theta_{\text{mlp}}^h$ that

$$f(x; \theta) = f_{\text{out}}(f_{\text{mlp}}(f_{\text{in}}(x); \text{proj}_{\text{mlp}}(\theta))). \quad (4)$$

Now, our goal is to test more general types of models. In particular, we generalize to an arbitrary alternative submodel $f_{\text{alt}} : \mathbb{R}^d \times \Theta_{\text{alt}} \to \mathbb{R}^d$ with $\text{proj}_{\text{alt}} : \Theta \to \Theta_{\text{alt}}$ such that

$$f(x; \theta) = f_{\text{out}}(f_{\text{alt}}(f_{\text{in}}(x); \text{proj}_{\text{alt}}(\theta))). \quad (5)$$

In order to test whether two models $\theta_1, \theta_2 \in \Theta$ of the more general form in equation (5) are independent, we will first construct proxy models of the form in equation (4) and then apply our previous test $\phi_{\text{MATCH}}$ to these proxy models. We construct these proxy models by leveraging the fact that $f_{\text{alt}}$ shares the same input and output space with $f_{\text{mlp}}$. Specifically, for $k \in \{1, 2\}$ we first learn parameters $\widehat{\theta}_k \in \Theta_{\text{mlp}}^h$ so that $f_{\text{mlp}}(\cdot ; \widehat{\theta}_k)$ approximates $f_{\text{alt}}(\cdot ; \text{proj}_{\text{alt}}(\theta_k))$. We then return $\phi_{\text{MATCH}}(\widehat{\theta}_1, \widehat{\theta}_2)$. We capture this two-stage process in Algorithm 5.

Perhaps surprisingly, we show that Algorithm 5 is effective in practice at distinguishing independent versus non independent models. The hidden dimension $h$ and input distribution

---

**Algorithm 5:** Generalized robust test
**Input:** Model parameters $\theta_1, \theta_2 \in \Theta$
**Parameters:** distribution $P$ over $\mathbb{R}^d$
**Output:** $\widehat{p} \in [0, 1]$
1 **for** $k \in \{1, 2\}$ **do**
2    $\widehat{\theta}_i \leftarrow$
    $\arg\min_{\widehat{\theta}} \mathbb{E}_{x \sim P} \left[ \left\| f_{\text{alt}}(x; \text{proj}_{\text{alt}}(\theta_k)) - f_{\text{mlp}}(x; \widehat{\theta}_k) \right\|^2 \right]$
3 **return** $\widehat{p} \leftarrow \phi_{\text{MATCH}}(\widehat{\theta}_1, \widehat{\theta}_2)$

---

$P$ with which we learn the GLU MLP are hyperparameters of the test. See Section 3.2 for details.

## 3. Experimental Results

### 3.1. Constrained setting

We first validate validate the effectiveness of our tests in the constrained setting on open-weight language models — 21 models trained with the Llama-7B architecture with public documentation on ground truth model independence. These models all contain $L = 32$ GLU MLPs, each part of its own Transformer block. We run experiments with three different tests. Each test comprises two elements: a test statistic along with a method for computing p-values from the statistic. For the first test, we use $\phi_{\ell_2}$ and compute p-values via PERMTEST with $T = 100$. For the other two tests, we compute p-values by directly aggregating the outputs of (respectively) $\phi_{U^{(\ell)}}$ and $\phi_{H^{(\ell)}}$ over $\ell \in [L]$ using FISHER. We obtain the inputs to the GLU MLP in the $\ell$-th required to compute $\phi_{H^{(\ell)}}$ by sampling sequences of tokens uniformly at random from the models' vocabulary and computing a forward pass through the full model while storing the MLP hidden layer activations. The equivariant transformation class $\Pi$ is the set of permutations over both the hidden units of each MLP (see Example 2) and the embedding dimension of the model (i.e., the inputs passed to the both the MLP and self-attention layers in each block); we defer the precise definition of $\Pi$ in this case to Appendix D.

#### 3.1.1. BASELINE STATISTICS

We employ two test statistics from prior work as baselines: Jensen-Shannon divergence between next token output distributions ($\phi_{\text{JSD}}$, (Lin, 2006)), and $\phi_{\ell_2}$ (Xu et al., 2024)) with PERMTEST (details in Section 3.2.2). We computed $\phi_{\text{JSD}}$ using input sequences sampled from WikiText-103 (Merity et al., 2017; Xu et al., 2024) (consistent with prior work). Since the Jensen-Shannon divergence is (by definition) invariant to any transformation of weights that does not affect model output, we cannot compute meaningful p-values using PERMTEST; instead, in our experiments we report the raw value of the test statistic itself.

### 3.1.2. LLAMA FAMILY EXPERIMENTAL RESULTS

The 21 models we evaluated include 6 base models (trained from scratch), so we have six disjoint sets of the models based on `Llama-2-7b-hf` stemming from a diverse mix of industry labs and non-profits (Azerbayev et al., 2024; Sudalairaj et al., 2024; Liu et al., 2024; Li et al., 2023). We consider any pair of models in the same tree as dependent and all other pairs as independent. We include examples of further fine-tunes (e.g., `llemma_7b`) of fine-tunes (e.g., `CodeLlama-7b-hf`) among the models we test. We will mostly refer to models using by their Huggingface identifiers, without the organization names for clarity.

We evaluated four test statistics: $\phi_{U^{(\ell)}}$ (cosine similarity of weights), $\phi_{H^{(\ell)}}$ (cosine similarity of hidden activations), $\phi_{\ell_2}$ ($\ell_2$ distance), and $\phi_{\mathrm{JSD}}$ (Jensen-Shannon Divergence). As we describe in Section 2.2.2, for $\phi_{U^{(\ell)}}$ and $\phi_{H^{(\ell)}}$ we report aggregated p-values over all blocks using `FISHER`. We report results for a subset of these pairs involving base model `Llama-2-7b-hf` in Table 1 while deferring the rest and the full experimental setup details to Appendix E.

| $\theta_1 =$ `Llama-2-7b-hf`, $\theta_2 =$? | Indep.? | $\phi_{\mathrm{JSD}}$ (log) | p-values |||
|---|---|---|---|---|---|
| | | | $\phi_{\ell_2}$ | $\phi_{U^{(\ell)}}$ | $\phi_{H^{(\ell)}}$ |
| `llama-7b-hf` | ✓ | -11.10 | 0.98 | 0.60 | 0.25 |
| `vicuna-7b-v1.1` | ✓ | -10.40 | 0.63 | 0.16 | 0.64 |
| `Amber` | ✓ | -10.69 | 0.75 | 0.36 | 0.88 |
| `open-llama-7b` | ✓ | -8.38 | 0.26 | 0.36 | 0.71 |
| `vicuna-7b-v1.5` | ✗ | -10.87 | 0.01 | $\varepsilon$ | $\varepsilon$ |
| `CodeLlama-7b-hf` | ✗ | -10.62 | 0.01 | $\varepsilon$ | $\varepsilon$ |
| `llemma-7b` | ✗ | -10.24 | 0.01 | $\varepsilon$ | $\varepsilon$ |
| `Orca-2-7b` | ✗ | -10.34 | 0.01 | $\varepsilon$ | $\varepsilon$ |

*Table 1.* We report various constrained setting test statistics with $\theta_1$ as `Llama-2-7b-hf` and $\theta_2$ ranging over the listed models. The "independent" column is the ground truth. Here, $\varepsilon = 2.2$e-308 (numerical underflow for a 64-bit float). We find our proposed tests $\phi_{U^{(\ell)}}$ and $\phi_{H^{(\ell)}}$ distinguish independent versus non-independent model pairs with high statistical power.

Consistent with prior work (Xu et al., 2024), we find that $\phi_{\mathrm{JSD}}$ does not reliably distinguish independent versus dependent model pairs. For example, `CodeLlama-7b-hf` exhibits a larger divergence with `Llama-2-7b-hf` than the independently-trained models `llama-7b-hf` and `Amber`.

All other test statistics reliably distinguish independent versus dependent pairs; in particular, the p-values we obtain using the other test statistics are negligible for all dependent pairs (for $\phi_{\ell_2}$, because we run `PERMTEST` with $T = 99$ for computational reasons, we cannot obtain a p-value less than 0.01. Notably, in contrast to our findings, prior work (Xu et al., 2024) argued that the $\ell_2$ distance between model parameters is not a reliable indicator of independence, in the sense that the $\ell_2$ distance between dependent pairs is sometimes larger than that of independent pairs (similar to the case of $\phi_{\mathrm{JSD}}$); the key difference is that Xu et al. (2024) report the raw $\ell_2$ distance whereas we obtain p-values from the raw distances using `PERMTEST`. We hypothesize that

`PERMTEST` effectively standardizes the raw distances. We further evaluated the efficacy of our tests through ablations by training two models with the same OLMo-7B architecture on the same dataset that only differ on the choice of random initialization of randomness, and report results in Appendix E.1. We also verify that Miqu-70B is not independent from Llama 2-70B (Mensch, 2024) and report further details in Appendix E.2.

## 3.2. Unconstrained setting

For the unconstrained setting, we first assess the previous 21 models of the Llama-7B architecture. We compute $\phi_{\mathrm{MATCH}}$ with the gate and up-projection matrices $M = H_{\mathrm{gate}}^\ell$ and $M' = H_{\mathrm{up}}^\ell$ of each MLP in block $\ell \in [L]$, and aggregate them with `FISHER`. We obtain the activations in the MLPs by using input sequences sampled from WikiText-103 and computing a forward pass through the full model, with results on all model pairs in Appendix F.

We find that the distribution of $\phi_{\mathrm{MATCH}}$ on independent model pairs is close to uniform (Figure 2), whereas across all non-independent model pairs the statistic is at most $\varepsilon$. Unlike the constrained setting, where the p-values are valid by construction, the output of the robust test does not enjoy such theoretical guarantees; however, Figure 2 suggests that even in the unconstrained setting our statistic $\phi_{\mathrm{MATCH}}$ behaves like a p-value.

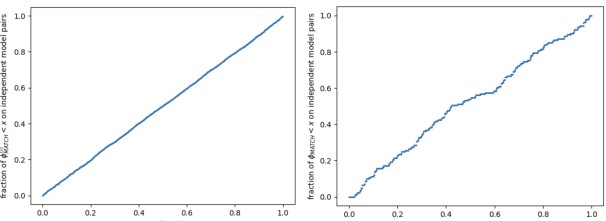

(a) Plot of $x \in [0, 1)$ vs. the fraction of $\phi_{\mathrm{MATCH}}^{(i)}$ (across all MLP blocks) of independent model pairs less than $x$.

(b) Plot of $x \in [0, 1)$ vs. the fraction of $\phi_{\mathrm{MATCH}}$ ($\phi_{\mathrm{MATCH}}^{(i)}$ aggregated with `FISHER`) of independent model pairs less than $x$.

*Figure 2.* We plot the fraction of $\phi_{\mathrm{MATCH}}$ less than $x \in [0, 1)$, aggregated with `FISHER` for independent model pairs. Both plots roughly follow the line $y = x$, i.e. a uniform distribution in $[0, 1)$ under the null, meaning $\phi_{\mathrm{MATCH}}$ empirically acts as a p-value.

We also validated our tests on the Mistral architecture—we compared the weights of the hybrid `StripedHyena-Nous-7B` (Poli et al., 2023) with `Mistral-7B-v0.1` and find non-independent parameters via $\phi_{U^{(\ell)}}$. We compute $\phi_{U^{(\ell)}}$ on all parameters, which allows us to identify non-independence between specific parameters of the models — such as the self-attention matrices — rather than as models as a whole, and report values of $\phi_{U^{(\ell)}}$ among certain parameters in Table 6 in

Appendix F.1. From the small p-values, we infer that the embedding layer and some self-attention matrices were likely shared between the two models.

### 3.2.1. SIMULATING STRONG-ISH ADVERSARIES

A significant difficulty in evaluating the robustness of our test $\phi_{\text{MATCH}}$ to adversarial transformations is that we cannot exhaustively enumerate all such transformations. Recalling that $\phi_{\text{MATCH}}$ specifically considers the MLP layers contained within two models, we attempt to fool it by randomly reinitializing and retraining these MLP layers individually, thus simulating a somewhat strong adversary.

We reinitialize the first GLU MLP module of a model $\theta_1$ with an MLP with double the width, and using Algorithm 5 (generalized robust test), we train $\hat{\theta_1}$ with random Gaussians as the training distribution $P$. We retrain each of the 32 MLPs (keeping other layers fixed) of `vicuna-7b-v1.5` (a finetune of `Llama-2-7b-hf`) for 10k gradient steps (until the loss curve plateaus). (Additional hyperparameters and a learning curve are in Appendix F.2.) For all 32 runs, we compute $\phi_{\text{MATCH}}$ for the retrained model with the original `Llama-2-7b-hf` and find $\phi_{\text{MATCH}}$ remains very small between the non-independent models even after an MLP has been retrained. For example, retraining the first MLP module, $\phi_{\text{MATCH}}^{(1)}$ on the first MLP was less than $\varepsilon = 2.2\text{e-}308$, indicating that the two models are not independent. We find the same is true for the other MLP layers as well (i.e. $\phi_{\text{MATCH}}^{(\ell)}$ when evaluated on retrained layer $\ell$), with full results in Table 7 of Appendix F.2.

### 3.2.2. GENERALIZING TO DIFFERENT ARCHITECTURES

As we describe in Section 2.3.2, we can also apply our test to model architectures which do not contain GLU MLP submodels. For example, the GPT-2 architecture uses a standard 2-layer MLP rather than a GLU MLP. We apply our test (Algorithm 5) to `GPT2_PMC` and `gpt2`, where the former is a finetune of the latter (Radford et al., 2019). We use 30k training steps with an isotropic Gaussian input distribution to learn the GLU MLP parameters with which we replace the original MLP submodels in each model. The test yields a value of 3.034e-61, thus distinguishing the two models as dependent. We show additional results on independent and non-independent models (of Llama and GPT architectures) in Appendix F.4.

### 3.3. Fine-grained forensics and Localized testing

Finally, we use $\phi_{\text{MATCH}}$ on models pairs with different dimensions, specifically on pruned model pairs, when model dimensions are reduced by preserving only select weights.

In particular, we were able to identify the specific Transformer blocks of `Llama-3.1-8B` whose weights were likely used in initializing `Llama-3.2-3B` and `Llama-3.2-1B`, as Meta reported that the first two models were pruned from the third (MetaAI, 2024). We match $\phi_{\text{MATCH}}^{(i,j)}$ with block $i$ from $\theta_1$ and $j$ from $\theta_2$, such that $\phi_{\text{MATCH}}^{(i,j)}$ is less than 1e-4. We report the matched layers between the `Llama-3.1` and `Llama-3.2` models in Figure 3 and in Appendix F.3.

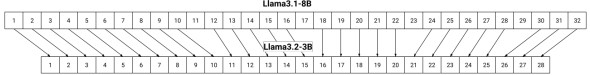

*Figure 3.* We evaluate $\phi_{\text{MATCH}}^{(i,j)}$ between all pairs of GLU MLPs of Llama 3.1-8B and Llama 3.2-3B. Arrows indicate if $\phi_{\text{MATCH}}^{(i,j)} <$ 1e-4 and suggest which Transformer blocks of Llama 3.1-8B were kept in the pruning process to initialize Llama 3.2-3B.

We also identify which hidden units were most likely shared between the blocks when MLP dimension is reduced (from 14336 to 8192) during pruning, from the permutation $\pi$ returned from the up projection matching, $\text{MATCH}(H_{\theta_1,\text{up}}^{(\ell)}, H_{\theta_2,\text{up}}^{(\ell)})$. We plot the activation matching for `Llama-3.1-8B` and `Llama-3.2-3B` in Appendix F.3.

## 4. Related & Future Work

A related line of work known as model fingerprinting (Xu et al., 2024; Zhang et al., 2025; Jin et al., 2024; Yang & Wu, 2024) plants a secret signal in the weights of a model so that anyone who knows the key can detect the fingerprint from query access to the model (or fine-tunes of the model). For example, Xu et al. (2024) propose fingerprinting a model by fine-tuning on a secret random string; fingerprint detection then resolves to prompting a putative fingerprinted model with a prefix of the string. Unlike Xu et al. (2024), we do not intervene on the training process of the models we test; however, we do require access to model weights.

Finally, a separate line of work on text watermarking aims to attribute model-generated text by planting a watermark when sampling text from the model (Christ et al., 2024; Kirchenbauer et al., 2023; Kuditipudi et al., 2024; Aaronson & Kirchner, 2023). Because it intervenes on sampling, text watermarking is inapplicable to open-weight models, the focus of both model fingerprinting and our setting. Recent work demonstrates that models can directly learn to generate watermarked text but also finds the learned watermark is not robust to further fine-tuning (Gu et al., 2024).

Future work can consider differentiating between fine-tunes of the same base model to reconstruct a complete "family tree" of model lineage is possible (e.g. infer Llemma is a direct fine-tune of CodeLlama) (Yax et al., 2025), and whether robustness against adversarial attacks is solvable with exact guarantees warrants further exploration.

## Acknowledgments

We gratefully acknowledge the support of this work by an NSF Frontier Award (NSF Grant no. 1805310) and Omidyar. Sally Zhu was supported by a Stanford CURIS Fellowship. Ahmed Ahmed is grateful to be supported by an NSF Graduate Research Fellowship and a Knight-Hennessy Fellowship.

## Impact statement

This paper presents work whose goal is to advance the field of Machine Learning. There are many potential societal consequences of our work, some of which we discuss in the introduction and none which we feel must be specifically highlighted here.

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

## A. Proofs of Main Theorems

*Proof of Theorem 1.* From our assumptions on $A$ and $P$ and the fact that $\{\pi_t\}_{t=1}^T$ are independently drawn, it follows that the collection $\{\pi_t(\theta_1)\}_{t=1}^T$ comprises $T$ exchangeable copies of $\theta_1$. The independence of $\theta_1$ and $\theta_2$ thus implies $\{(\pi_t(\theta_1), \theta_2)\}_{t=1}^T$ comprises $T$ exchangeable copies of $(\theta_1, \theta_2)$, and so the claim follows by symmetry — $\phi(\theta_1, \theta_2)$ is identically distributed as $\{\phi(\pi_t(\theta_1), \theta_2)\}_{t=1}^T$, so $\phi(\theta_1, \theta_2)$ will have uniform rank among the other values. Ties ($\phi_t = \phi(\theta_1, \theta_2)$) randomly contribute to $\hat{p}$, so symmetry still holds and the p-values will be uniformly distributed under the null. □

*Proof of Theorem 2.* As $M$ is a $\Pi$-equivariant map, if $\theta_1 \perp \theta_2$ then letting $\pi = \texttt{LAP}(C)$ in $\texttt{MATCH}$ is equivalent in distribution to sampling $\pi \sim \text{Unif}(\Pi)$. Then the output of $\texttt{MATCH}$ is identical in distribution for *any* pair of independent models, and can be converted to a p-value using $\texttt{SPEARMAN}$ and the distribution for the Spearman correlation coefficient (t-distribution with $h - 2$ degrees of freedom). □

*Proof of Theorem 3.* Let $\theta_1' \sim A(\pi_1^{(i)} \circ \pi_2^{(j)}(\theta_1^0))$ for $\pi_1, \pi_2 \overset{\text{i.i.d.}}{\sim} \text{Unif}(\Pi)$. Then $\theta_1'$ is an independent copy of $\theta_1$ since taking the composition $\pi_1^{(i)} \circ \pi_2^{(j)}(\theta_1)$ yields an independent copy of $\theta_1$ for any $\pi_1, \pi_2 \in \Pi$. From $\theta_1 \perp \theta_2$, it follows for $\ell \in \{i, j\}$ that $\texttt{MATCH}(M^{(\ell)}(\theta_1'), M^{(\ell)}(\theta_2))$ is identically distributed to $\texttt{MATCH}(M^{(\ell)}(\theta_1), M^{(\ell)}(\theta_2))$. The result then follows from the fact $\texttt{MATCH}$ is equivariant with respect to permuting the rows of its arguments: in particular, for any $\pi \in \Pi$ we have $\texttt{MATCH}(\pi W_1, W_2) = \pi \texttt{MATCH}(W_1, W_2)$. □

## B. Randomized Learning Algorithms

One notable (non-contrived) category of deep learning algorithms that are *not* permutation equivariant are those with random dropout masks to hidden units during training. In particular, once we fix a specific setting of mask values to specify a deterministic learning algorithm, this algorithm will not be permutation equivariant unless the individual dropout masks are all permutation invariant (which is highly unlikely). We provide a generalized statement of Theorem 1 for randomized algorithms.

**Definition 5.** Let $\Pi \subset \Theta \to \Theta$. Let $\pi \in \Pi$ and $\theta^0 \in \Theta$, with $\bar{\theta} \sim A(\theta^0), \theta = \pi(\bar{\theta})$ and $\theta' \sim A(\pi(\theta_0))$. A randomized learning algorithm $A : \Theta \to \mathcal{P}(\Theta)$ is $\Pi$-*equivariant* if and only if $\theta \overset{d}{=} \theta'$.

**Theorem 4.** *Let $\phi : \Theta \times \Theta \to \mathbb{R}$ be a test statistic and $\Pi \subset \Theta \to \Theta$ be finite. Let $A : \Theta \to \mathcal{P}(\Theta)$ be $\Pi$-equivariant and let $P \in \mathcal{P}(\Theta)$ be $\Pi$-invariant. Let $\theta_1, \theta_2 \in \Theta$ be independent random variables, with $\theta_1 \sim A(\theta_1^0)$ for $\theta_1^0 \sim P_1$. Then $\widehat{p} = \texttt{PERMTEST}(\theta_1, \theta_2)$ is uniformly distributed on $\{\frac{i}{T+1}\}_{i=1}^T$.*

*Proof.* The proof is identical to that of Theorem 1. □

## C. Transformer Architecture and Notation

We consider models with the Llama Transformers architecture and define the notation henceforth, although this can easily be extended to other Transformer architectures.

Following the definition of $f_{\text{mlp}}$ in Example 1, we can define an abstraction of the full Llama language model architecture consisting of $L$ Transformer blocks sandwiched between an input and output layer. For the sequel, we will abuse notation in applying $f_{\text{mlp}}$ to multi-dimensional tensors by broadcasting along the last axis. We use $d, n \in \mathbb{N}$ to respectively denote the model dimension and sequence length, where $\Theta_{\text{LM}} = \Theta_{\text{in}} \times \Theta_{\text{block}}^{\times L} \times \Theta_{\text{out}}$ with $\Theta_{\text{block}}$ denoting the parameter space of each Transformer block and $\Theta_{\text{in}}, \Theta_{\text{out}}$ denoting the parameter spaces the input and output layers. We decompose $\Theta_{\text{block}} = \Theta_{\text{attn}} \times \Theta_{\text{mlp}}$ and use $f_{\text{rest}} : \Theta_{\text{attn}} \times \mathbb{R}^{n \times d} \to \mathbb{R}^{n \times d}$ to denote all remaining parts of the Transformer besides the MLP. The inputs to $f_{\text{rest}}$ are the input and output of the MLP, and the output of $f_{\text{rest}}$ is fed directly to the MLP of the next layer. In particular, $f_{\text{rest}}$ takes the input and output to the MLP of layer $i$, and first performs the residual connection following the MLP of layer $i$, then the self-attention and normalization components of layer $i + 1$, and returns the input to the MLP of layer $i + 1$. We use $f_{\text{in}} : \Theta_{\text{in}} \times \mathcal{X} \to \mathbb{R}^{n \times d}$ and $f_{\text{out}} : \Theta_{\text{block}}^{(L)} \times \mathbb{R}^{n \times d} \to \mathcal{Y}$ to respectively denote the input and output layers, i.e. the elements before the first MLP and after the last MLP. Putting everything together gives the following definition of the model; we introduce the notation $X_\theta^{(i)}$ in the definition as a matter of convenience to track intermediate activations.

**Definition 6.** (GLU Transformer model) Let $\theta = (\theta_{\text{in}}, \{\theta_{\text{block}}^{(i)}\}_{i=1}^L, \theta_{\text{out}}) \in \Theta_{\text{LM}}$ and $X \in \mathcal{X}$, with $\theta_{\text{block}}^{(i)} = (\theta_{\text{attn}}^{(i)}, \theta_{\text{mlp}}^{(i)})$. Then $f_{\text{LM}}(X; \theta) = f_{\text{out}}(X_\theta^{(L)}; \theta_{\text{out}})$ for $X_\theta^{(0)} = f_{\text{in}}(X; \theta_{\text{in}})$ and

$$X_\theta^{(i)} = f_{\text{rest}}(X_\theta^{(i-1)}, f_{\text{mlp}}(X_\theta^{(i-1)})). \tag{6}$$

For a Llama model, table 2 describes the shapes of the model weight matrices for $i = 1, \ldots, L$, for $V$ (vocab size), $d_{\text{emb}}$ (the hidden dimension), and $d_{\text{mlp}}$ (MLP hidden dimension). Following Definition 6, we have $\theta_{\text{in}} = (E), \theta_{\text{block}}^{(i)} = (\theta_{\text{attn}}^{(i)}, \theta_{\text{mlp}}^{(i)})$ where

| Parameter name | Notation |
|---|---|
| embedding | $E \in \mathbb{R}^{V \times d_{\text{emb}}}$ |
| input layernorm | $\gamma_{\text{input},i} \in \mathbb{R}^{1 \times d_{\text{emb}}}$ |
| attention query matrix | $W_{Q,i} \in \mathbb{R}^{d_{\text{emb}} \times d_{\text{emb}}}$ |
| attention key matrix | $W_{K,i} \in \mathbb{R}^{d_{\text{emb}} \times d_{\text{emb}}}$ |
| attention value matrix | $W_{V,i} \in \mathbb{R}^{d_{\text{emb}} \times d_{\text{emb}}}$ |
| attention output matrix | $W_{O,i} \in \mathbb{R}^{d_{\text{emb}} \times d_{\text{emb}}}$ |
| post-attention layernorm | $\gamma_{\text{post-attn},i} \in \mathbb{R}^{1 \times d_{\text{emb}}}$ |
| MLP gate projection | $G_i \in \mathbb{R}^{d_{\text{mlp}} \times d_{\text{emb}}}$ |
| MLP up projection | $U_i \in \mathbb{R}^{d_{\text{mlp}} \times d_{\text{emb}}}$ |
| MLP down projection | $D_i \in \mathbb{R}^{d_{\text{emb}} \times d_{\text{mlp}}}$ |
| final layernorm | $\gamma_{\text{final}} \in \mathbb{R}^{1 \times d_{\text{emb}}}$ |
| linear output | $O \in \mathbb{R}^{d_{\text{emb}} \times V}$ |

*Table 2.* Llama model architecture and dimensions.

$\theta_{\text{attn}}^{(i)} = (\gamma_{\text{input},i}, W_{Q,i}, W_{K,i}, W_{V,i}, W_{O,i}, \gamma_{\text{post-attn}}^{(i)})$, $\theta_{\text{mlp}}^{(i)} = (G_i, U_i, D_i)$, and $\theta_{\text{out}} = (\gamma_{\text{final}}, L)$. We now describe a forward pass of the model.

We define the softmax function on a vector $v = (v_1, \ldots, v_n)$, softmax$(v)$, as

$$\text{softmax}(v)_i = \frac{e^{v_i}}{\sum_{k=1}^{n} e^{v_k}}.$$

On batched input $X \in \mathbb{R}^{N \times n \times m}$ where each $X^{(b)} = [w_1 | \ldots | w_m] \in \mathbb{R}^{n \times m}$ with column vectors $w_i$, we define the softmax as

$$\text{softmax}(X^{(b)}) = [\text{softmax}(w_1) | \ldots | \text{softmax}(w_m)],$$

$$\text{softmax}(X) = [\text{softmax}(X^{(1)}) | \ldots | \text{softmax}(X^{(N)})].$$

For a forward pass of the model $f_{\text{LM}}(X; \theta)$, consider an input sequence of tokens $X \in \{0, 1\}^{N \times V}$ as one-hot vectors where $n$ is sequence length. Then

We feed the input through:

1. ($f_{\text{in}}$) Embedding layer:
$$X_\theta^{(0)} = f_{\text{in}}(X; \theta_{\text{in}}) = XE \in \mathbb{R}^{N \times d_{\text{emb}}}$$

2. ($f_{\text{attn}}, f_{\text{mlp}}, f_{\text{post}}$) For each Transformer block $i = 0, 1, \ldots, L$, through $f_{\text{attn}}$, $f_{\text{mlp}}$, and $f_{\text{post}}$:

   (a) Input layernorm:
   $$X_{\text{LN}_1}^{(i)} = \frac{X_\theta^{(i)}}{\sqrt{\text{Var}(X_\theta^{(i)}) + \varepsilon}} \odot \gamma_{\text{input},i}$$

   (with variance over the last axis) for some offset $\varepsilon$ (typically 1e-6).

   (b) Causal multi-head self-attention: Split $X_{\text{LN}_1}^{(i)}$ on the first axis into nheads $X_{\text{LN}_1,j}^{(i)}, \ldots, X_{\text{LN}_1,\text{nheads}}^{(i)}$. On each head $X_{\text{LN}_1,j}^{(i)}$,

   $$X_{\text{SA},j}^{(i)} = \text{self-attn}(X_{\text{LN}_1,j}^{(i)}) = \text{softmax}\left( \frac{X_{\text{LN}_1,j}^{(i)} W_{Q,i}^T (X_{\text{LN}_1,j}^{(i)} W_{K,i}^T)^T}{\sqrt{d_{\text{emb}}}} \right) X_{\text{LN}_1,j}^{(i)} W_{V,i}^T W_{O,i}^T$$

   and concatenate $X_{\text{SA},j}^{(i)}$ along the first axis again as $X_{\text{SA}}^{(i)}$.

   (c) Dropout and residual connection: $X_{\text{DR}_1}^{(i)} = X_{\text{LN}_1}^{(i)} + \text{Dropout}(X_{\text{SA}}^{(i)})$

   (d) Post-attention layernorm:
   $$X_{\text{LN}_2}^{(i)} = \frac{X_{\text{DR}_1}^{(i)}}{\sqrt{\text{Var}(X_{\text{DR}_1}^{(i)}) + \varepsilon}} \odot \gamma_{\text{post-attn},i}$$

   (with variance over the last axis) for some offset $\varepsilon$. Then we have

   $$f_{\text{attn}}(X_\theta^{(i-1)}; \theta_{\text{attn}}^{(i)}) = X_{\text{LN}_2}^{(i)}.$$

| Parameter name | $\theta$ | $\pi_{\text{emb}}(\theta)$ | $\pi_{\text{mlp}}(\theta)$ |
|---|---|---|---|
| embedding | $E$ | $E\pi_{\text{emb}}$ | $E$ |
| input layernorm | $\gamma_{\text{input},i}$ | $\gamma_{\text{input},i}\pi_{\text{emb}}$ | $\gamma_{\text{input},i}$ |
| attention query matrix | $W_{Q,i}$ | $W_{Q,i}\pi_{\text{emb}}$ | $W_{Q,i}$ |
| attention key matrix | $W_{K,i}$ | $W_{K,i}\pi_{\text{emb}}$ | $W_{K,i}$ |
| attention value matrix | $W_{V,i}$ | $W_{V,i}\pi_{\text{emb}}$ | $W_{V,i}$ |
| attention output matrix | $W_{O,i}$ | $\pi_{\text{emb}}^T W_{O,i}$ | $W_{O,i}$ |
| post-attention layernorm | $\gamma_{\text{post-attn},\,i}$ | $\gamma_{\text{post-attn},\,i}\pi_{\text{emb}}$ | $\gamma_{\text{post-attn},\,i}$ |
| MLP gate projection | $G_i$ | $G_i\pi_{\text{emb}}$ | $\pi_{\text{mlp},i}G_i$ |
| MLP up projection | $U_i$ | $U_i\pi_{\text{emb}}$ | $\pi_{\text{mlp},i}U_i$ |
| MLP down projection | $D_i$ | $\pi_{\text{emb}}^T D_i$ | $D_i\pi_{\text{mlp},i}^T$ |
| final layernorm | $\gamma_{\text{final}}$ | $\gamma_{\text{final}}\pi_{\text{emb}}$ | $\gamma_{\text{final}}$ |
| linear output | $O$ | $\pi_{\text{emb}}^T O$ | $O$ |

*Table 3.* Transformations $\pi_{\text{emb}}$ and $\pi_{\text{mlp}}$ applied to a Llama-architecture model.

(e) Next, we feed through $f_{\text{mlp}}$, the multi-layer perceptron:

$$f_{\text{mlp}}(X_{\text{LN}_2}^{(i)}; \theta_{\text{mlp}}^{(i)}) = X_i^{\text{MLP}} = [\sigma(X_i^{\text{LN}_2}G_i^T) \odot (X_i^{\text{LN}_2}U_i^T)]D_i^T$$

for some activation $\sigma$ (e.g., SiLU).

(f) Finally, we feed through $f_{\text{post}}$, dropout and the residual connection:

$$f_{\text{post}}(\theta_{\text{mlp}}^{(i)}) = X_\theta^{(i+1)} = X_i^{\text{DR}_1} + \text{Dropout}(X_i^{\text{MLP}})$$

3. ($f_{\text{out}}$) Final layernorm on the output $X_\theta^{(N+1)}$ from the final Transformer block:

$$X_{\text{LN}}^{(L)} = \frac{X_\theta^{(L)}}{\sqrt{\text{Var}(X_\theta^{(L)}) + \varepsilon}} \odot \gamma_{\text{final}}$$

(with variance over the last axis) for some offset $\varepsilon$. Then, linear output embedding and softmax mapping to output probabilities:

$$f_{\text{out}}(X_\theta^{(L)}) = \text{softmax}(X_{\text{LN}}^{(L)}O^T),$$

which defines the entire forward pass $f_{\text{LM}}(X; \theta)$.

# D. Model Transformation Class

We describe two sets of equivariant transformations $\Pi$ on a Transformer model as described in Appendix C. (Abusing notation), the first set, $\Pi_{\text{emb}}$, consists of elements $\pi_{\text{emb}}$ where $\pi_{\text{emb}} \in \mathbb{R}^{d_{\text{emb}} \times d_{\text{emb}}}$ is a permutation matrix. The second set, $\Pi_{\text{mlp}}$, consists of elements $\pi_{\text{mlp}}$ where $\pi_{\text{mlp}} \in \mathbb{R}^{d_{\text{mlp}} \times d_{\text{mlp}}}$ is a permutation matrix.

1. $\pi_{\text{emb}}(\theta)$: Applying an embedding permutation $\pi_{\text{emb}} \in \mathbb{R}^{d_{\text{emb}} \times d_{\text{emb}}}$ by left or right multiplying all relevant matrices by $\xi_{\text{embed}}$ (permuting rows or columns).

2. $\pi_{\text{mlp}}(\theta)$: Applying MLP permutations $\pi_{\text{mlp},i} \in \mathbb{R}^{d_{\text{mlp}} \times d_{\text{mlp}}}$ to MLP layers.

These permutations are applied such that the outputs of the original model $\theta$ and the permuted model $\Pi(\theta)$ remain aligned. We describe the details in Table 3.

# E. Additional Constrained Setting Experimental Results

We report p-values from the statistics $\phi_{\ell_2}$, $\phi_{U^{(\ell)}}$, and $\phi_{H^{(\ell)}}$ on all 210 model pairs (from 21 Llama 2-architecture models) in Figures 4, 5, and 6, where the model names are colored by base model (ground truth). For all statistics, the p-values on independent model pairs are uniformly distributed, while they are all significant at 0.01 (smaller for $\phi_{U^{(\ell)}}$ and $\phi_{H^{(\ell)}}$) for fine-tuned model pairs.

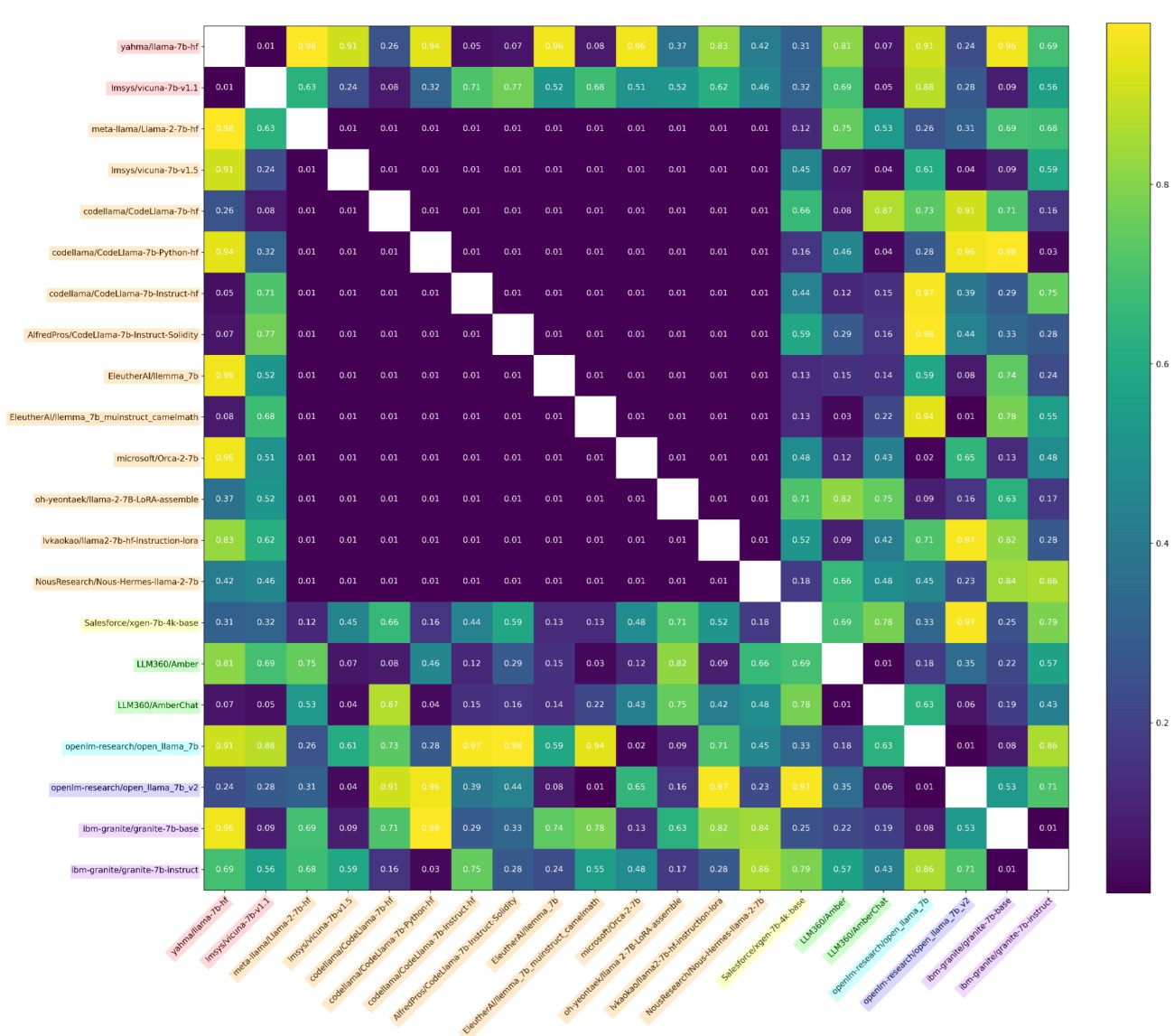

*Figure 4.* Results of p-values from $\phi_{\ell_2}$ on all model pairs.

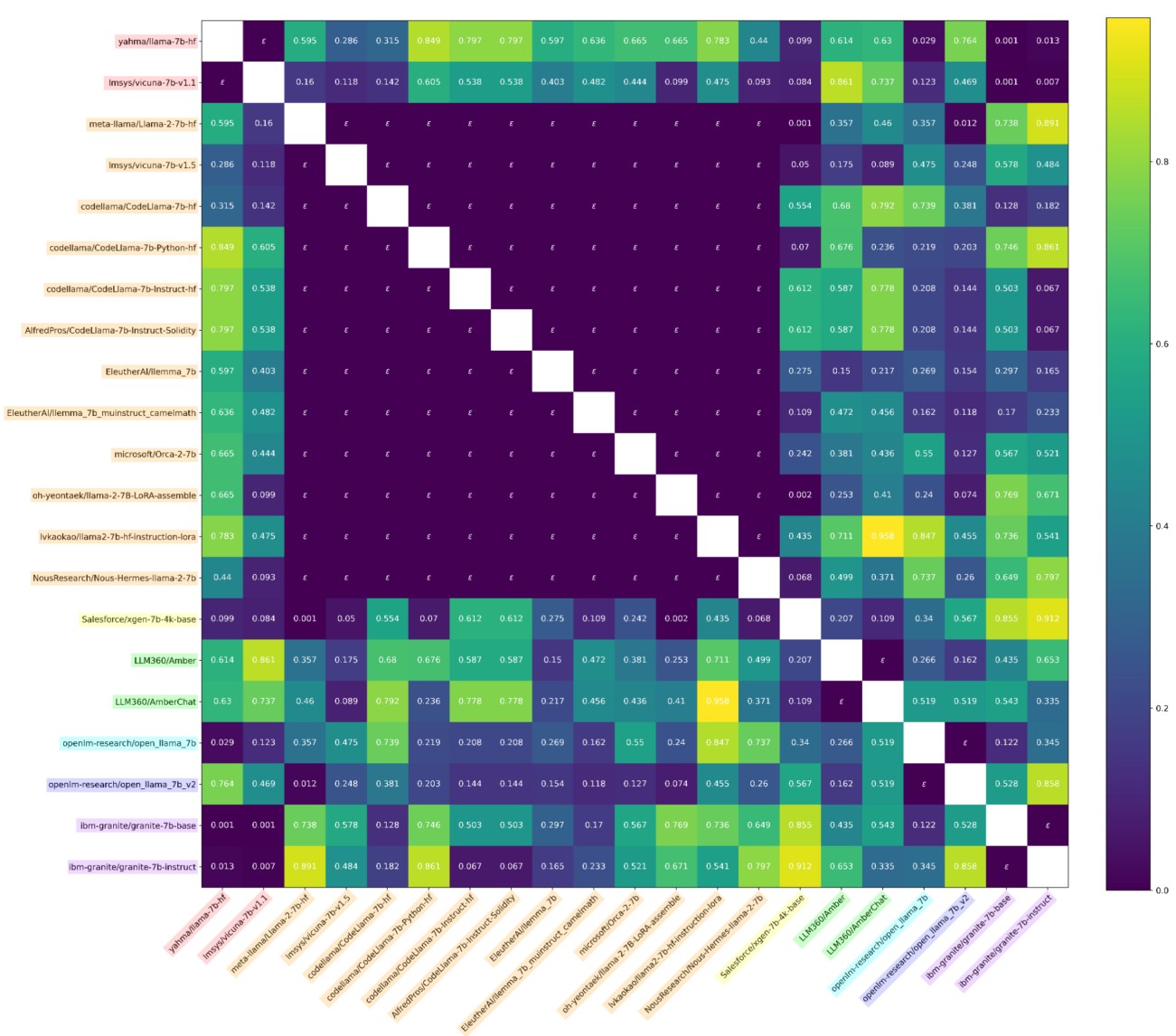

*Figure 5.* Results of p-values from $\phi_{U(\ell)}$ on all model pairs, where $\varepsilon = 2.2\text{e-}308$.

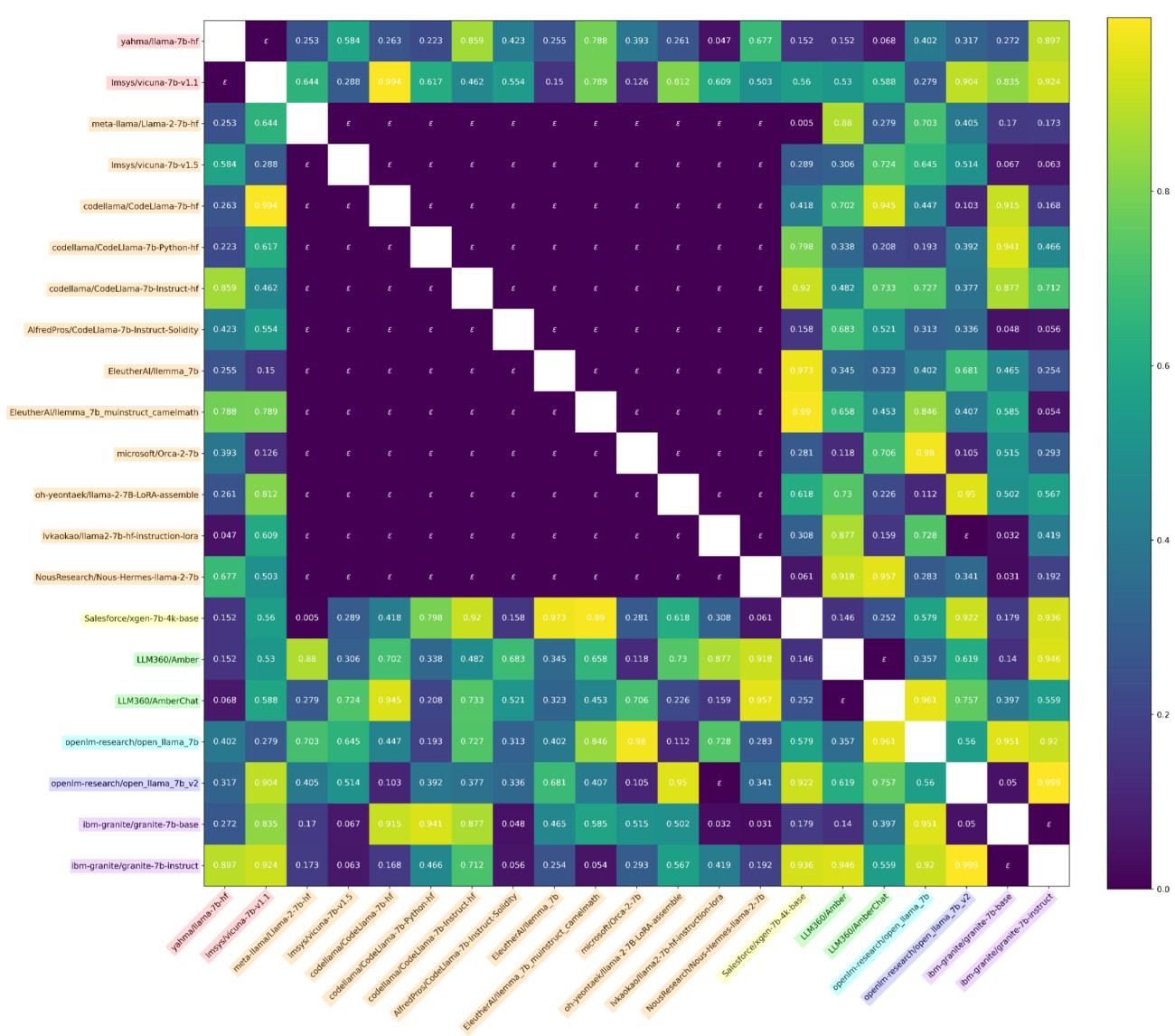

*Figure 6.* Results of p-values from $\phi_{H^{(\ell)}}$ on all model pairs, where $\varepsilon = $ 2.2e-308.

| # train tokens | $\phi_{U(\ell)}$ | $\phi_{H(\ell)}$ | $\phi_{\ell_2}$ | $\phi_{\text{MATCH}}$ | $\phi_{\text{JSD}}$ (log) |
|---|---|---|---|---|---|
| 100M | 0.641 | 0.119 | 0.07 | 0.809 | -11.81 |
| 1B | 0.789 | 0.483 | 0.06 | 0.443 | -11.05 |
| 10B | 0.707 | 0.277 | 0.93 | 0.343 | -11.28 |
| 18B | 0.819 | 0.141 | 0.64 | 0.027 | -11.03 |

*Table 4.* Results for $\phi_{U(\ell)}, \phi_{H(\ell)}$, and $\phi_{\text{MATCH}}$ evaluated on training checkpoints between two independently-trained OLMo models.

| $\theta_1 = $ Llama-2-70b-hf, $\theta_2 = $ | $\phi_{U(\ell)}$ |
|---|---|
| miqu-1-70b-pytorch | $\varepsilon$ |
| Llama-3.1-70B | 0.571 |
| Palmyra-Fin-70B-32K | 0.539 |

*Table 5.* Results of $\phi_{U(\ell)}$ (aggregated with FISHER) with $\theta_1$ as Llama-2-70b-hf and $\theta_2$ ranging over the listed models.

### E.1. Identically distributed, Independent models

We further evaluated the efficacy of our tests through ablations by training two models with the same architecture on the same dataset that only differ on the choice of random initialization of randomness. Specifically, we ensure that our test does not incorrectly detect two similar (trained using the same learning algorithm) but independent (randomly initialized) models, as non-independent.

To verify this, we randomly initialized a model with the OLMo (7B) architecture (Groeneveld et al., 2024) and trained it on the Dolma v1_7 dataset ((Soldaini et al., 2024)). We trained a second model with independently chosen initialization and data ordering.

We keep checkpoints for both seeds after 100M, 1B, 10B, and 18B train tokens and evaluate the statistics $\phi_{U(\ell)}, \phi_{H(\ell)}$, and $\phi_{\text{MATCH}}$ on the two models at each training checkpoint, reported in Table 4. We highlight that the p-values are broadly distributed, validating our tests support independence even on two similarly-trained but independent models.

### E.2. Tests for Larger Models

Next, we evaluated our tests on larger models. We ran $\phi_{U(\ell)}$ on four 70B parameter models with the Llama 2-70B architecture shown in Table 5, and in particular, we verify that Miqu-70B is not independent from Llama 2-70B.

## F. Additional Unconstrained Setting Experimental Results

We report values of $\phi_{\text{MATCH}}$ on all model pairs in Figure 7. The statistic is low ($< \varepsilon = 10^{-308}$) for all non-independent model pairs, and uniformly distributed for independent model pairs, empirically acting as a p-value.

### F.1. Striped Hyena Experiments

We report $\phi_{U(\ell)}$ on specific parameters from StripedHyena-Nous-7B and Mistral-7B-v0.1 shown in Table 6. We no longer only evaluate $\phi_{U(\ell)}$ on MLP up projection matrices, so that we can investigate similarity in other parameters as well. These p-values no longer satisfy the independence requirement of Theorem 2, so we do not aggregate them with FISHER.

| Parameter name | Notation | $\phi_{U(\ell)}$ |
|---|---|---|
| embedding | $E$ | 1.61e-16 |
| attention query matrix | $W_Q^{(1)}$ | 6.17e-190 |
| attention key matrix | $W_K^{(1)}$ | 1.47e-7 |
| attention value matrix | $W_V^{(1)}$ | 1.56e-114 |
| attention query matrix | $W_Q^{(1)}$ | 6.17e-190 |
| attention output matrix | $W_O^{(1)}$ | 0.010 |
| MLP gate projection | $G^{(1)}$ | 0.517 |
| MLP up projection | $U^{(1)}$ | 0.716 |
| MLP down projection | $D^{(1)}$ | 6.03e-80 |

*Table 6.* $\phi_{U(\ell)}$ on parameters from StripedHyena-Nous-7B and Mistral-7B-v0.1, some with low p-values.

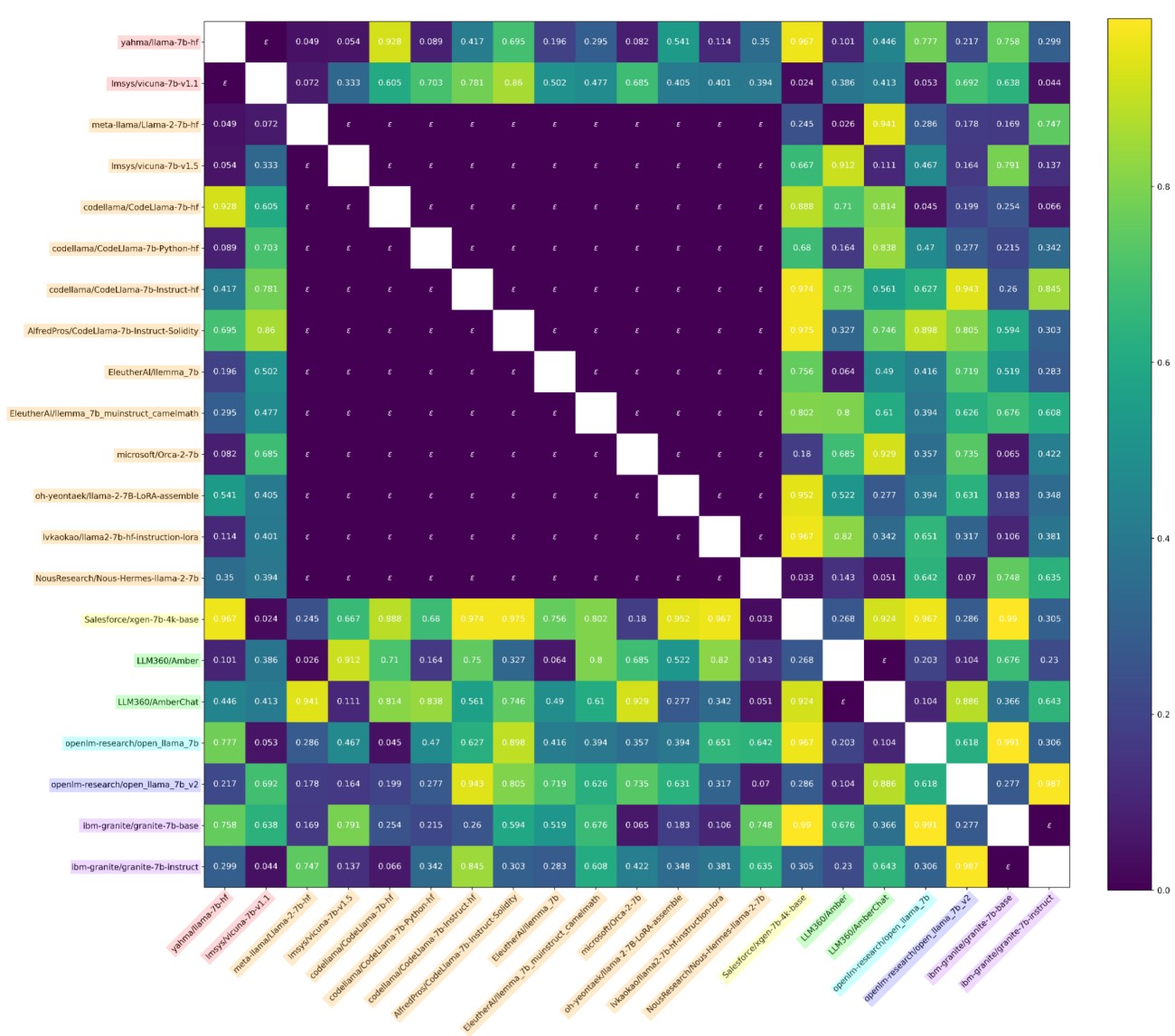

*Figure 7.* Results of values of $\phi_{\text{MATCH}}$ on all model pairs, where $\varepsilon = 2.2\text{e-}308$.

| MLP | Loss | $\log_{10}(\phi^{(i)}_{\text{MATCH}})$ | MLP | Loss | $\log_{10}(\phi^{(i)}_{\text{MATCH}})$ | MLP | Loss | $\log_{10}(\phi^{(i)}_{\text{MATCH}})$ |
|---|---|---|---|---|---|---|---|---|
| 1 | 0.0048 | $-479$ | 12 | 0.0060 | $-342$ | 23 | 0.0043 | $-593$ |
| 2 | 0.012 | $-485$ | 13 | 0.0058 | $-330$ | 24 | 0.0047 | $-542$ |
| 3 | 0.0026 | $-614$ | 14 | 0.0066 | $-323$ | 25 | 0.0050 | $-497$ |
| 4 | 0.0034 | $-580$ | 15 | 0.0063 | $-414$ | 26 | 0.0051 | $-534$ |
| 5 | 0.0030 | $-523$ | 16 | 0.0061 | $-394$ | 27 | 0.0052 | $-482$ |
| 6 | 0.0035 | $-513$ | 17 | 0.0063 | $-445$ | 28 | 0.0061 | $-477$ |
| 7 | 0.0041 | $-533$ | 18 | 0.0055 | $-515$ | 29 | 0.0065 | $-433$ |
| 8 | 0.0042 | $-464$ | 19 | 0.0045 | $-571$ | 30 | 0.0098 | $-361$ |
| 9 | 0.0050 | $-439$ | 20 | 0.0045 | $-512$ | 31 | 2.313 | $-26.4$ |
| 10 | 0.0050 | $-377$ | 21 | 0.0047 | $-595$ | 32 | 0.0114 | $-174$ |
| 11 | 0.0060 | $-365$ | 22 | 0.0043 | $-555$ | | | |

*Table 7.* $\phi_{\text{MATCH}}$ on individual blocks between `Llama-2-7b-hf` and `vicuna-7b-v1.5` after retraining MLP layers.

## F.2. MLP Retraining Experiments

We retrain each of the 32 MLP layers by feeding in random inputs through the original MLP (gate, up, and down projection matrices.) We train for 10000 gradient steps using MSE loss and an Adam Optimizer with a learning rate of 0.001 and batch size of 5000. A sample learning curve is in Figure 8.

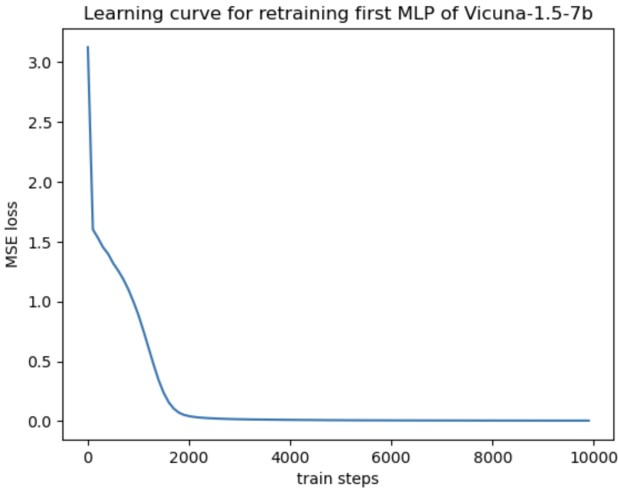

*Figure 8.* Learning curve for MLP retraining.

The MLP retraining results for all 32 MLP layers of `vicuna-7b-v1.5`, compared with `Llama-2-7b-hf` are in Table 7, showing that the statistic is robust to retraining of all layers.

## F.3. Localized Testing

As described in 4.4.2, we can run $\phi_{\text{MATCH}}$ on all pairs of Transformer blocks between two models (of different architecture), as long as they share the GLU structure. In addition to the Llama 3 results, we report results of matched blocks on the Sheared-LLaMa and Nvidia-Minitron models, which are both pruned from Llama models.

In particular, we were able to identify the specific Transformer blocks of $\theta_{8B} = $ `Llama-3.1-8B` whose weights were likely used in initializing $\theta_{3B} = $ `Llama-3.2-3B` and $\theta_{1B} = $ `Llama-3.2-1B`, as Meta reported that the `Llama-3.2-3B` and `Llama-3.2-1B` models were pruned from `Llama-3.1-8B` ((MetaAI, 2024)). We use $\phi_{\text{MATCH}}$ on all pairs of MLP blocks, where $(d_{\theta_{8B}}, h_{\theta_{8B}}, N_{\theta_{8B}}) = (4096, 14336, 32), (d_{\theta_{3B}}, h_{\theta_{3B}}, N_{\theta_{3B}}) = (3072, 8192, 28)$, and $(d_{\theta_{1B}}, h_{\theta_{1B}}, N_{\theta_{1B}}) = (2048, 8192, 16)$. We match blocks when the statistic $\phi^{(i,j)}_{\text{MATCH}}$ from block $i$ of model 1 and block $j$ of model 2 is less than 1e-4, reported in Tables 8 and 9 (with the same for the other matchings in this section).

| $i$ | 1 | 2 | 3 | 4 | 5 | 6 | 7 | 8 | 9 | 10 | 11 | 12 | 13 | 14 | 15 | 16 |
|---|---|---|---|---|---|---|---|---|---|---|---|---|---|---|---|---|
| $j : \phi_{\text{MATCH}}^{(i,j)}(\theta_{8B}, \theta_{3B}) < 1e-4$ | 1 | 2 | 3 | 4 | 5 | 6 | 7 | 8 | 9 | 10 | | 11 | 12 | 13 | 14 | 15 |

| $i$ | 17 | 18 | 19 | 20 | 21 | 22 | 23 | 24 | 25 | 26 | 27 | 28 | 29 | 30 | 31 | 32 |
|---|---|---|---|---|---|---|---|---|---|---|---|---|---|---|---|---|
| $j : \phi_{\text{MATCH}}^{(i,j)}(\theta_{8B}, \theta_{3B}) < 1e-4$ | | 16 | 17 | 18 | 19 | 20 | | 21 | 22 | 23 | 24 | 25 | | 26 | 27 | 28 |

*Table 8.* $\theta_{8B} = $ `Llama-3.1-8B` blocks matched with $\theta_{3B} = $ `Llama-3.2-3B` blocks using $\phi_{\text{MATCH}}$

| $i$ | 1 | 2 | 3 | 4 | 5 | 6 | 7 | 8 | 9 | 10 | 11 | 12 | 13 | 14 | 15 | 16 |
|---|---|---|---|---|---|---|---|---|---|---|---|---|---|---|---|---|
| $j : \phi_{\text{MATCH}}^{(i,j)}(\theta_{8B}, \theta_{1B}) < 1e-4$ | 1 | 2 | 3 | 4 | 5 | 6 | | | 7 | | | 8 | | | 9 | |

| $i$ | 17 | 18 | 19 | 20 | 21 | 22 | 23 | 24 | 25 | 26 | 27 | 28 | 29 | 30 | 31 | 32 |
|---|---|---|---|---|---|---|---|---|---|---|---|---|---|---|---|---|
| $j : \phi_{\text{MATCH}}^{(i,j)}(\theta_{8B}, \theta_{1B}) < 1e-4$ | | 10 | | | 11 | | | | | | | | | | 15 | 16 |

*Table 9.* $\theta_{8B} = $ `Llama-3.1-8B` blocks matched with $\theta_{1B} = $ `Llama-3.2-1B` blocks using $\phi_{\text{MATCH}}$

Next, we have Sheared-LLaMa 2.7B, with 32 Transformer blocks, hidden dimension 2560 and MLP dimension 6912. All 32 blocks align with the 32 blocks of Llama 2 7B, although both hidden and MLP dimensions have been reduced through pruning.

| $i$ | 1 | 2 | 3 | 4 | 5 | 6 | 7 | 8 | 9 | 10 | 11 | 12 | 13 | 14 | 15 | 16 |
|---|---|---|---|---|---|---|---|---|---|---|---|---|---|---|---|---|
| $j : \phi_{\text{MATCH}}^{(i,j)}(\theta_1, \theta_2) < 1e-90$ | 1 | 2 | 3 | 4 | 5 | 6 | 7 | 8 | 9 | 10 | 11 | 12 | 13 | 14 | 15 | 16 |

| $i$ | 17 | 18 | 19 | 20 | 21 | 22 | 23 | 24 | 25 | 26 | 27 | 28 | 29 | 30 | 31 | 32 |
|---|---|---|---|---|---|---|---|---|---|---|---|---|---|---|---|---|
| $j : \phi_{\text{MATCH}}^{(i,j)}(\theta_1, \theta_2) < 1e-90$ | 17 | 18 | 19 | 20 | 21 | 22 | 23 | 24 | 25 | 26 | 27 | 28 | 29 | 30 | 31 | 32 |

*Table 10.* $\theta_1 = $ `Sheared-LLaMa 1.3B` blocks matched with $\theta_2 = $ `Llama-2-7B` blocks using $\phi_{\text{MATCH}}$

Next, we have Sheared-LLaMa 1.3B, with 24 Transformer blocks, hidden dimension 2048 and MLP dimension 5504.

| $i$ | 1 | 2 | 3 | 4 | 5 | 6 | 7 | 8 | 9 | 10 | 11 | 12 |
|---|---|---|---|---|---|---|---|---|---|---|---|---|
| $j : \phi_{\text{MATCH}}^{(i,j)}(\theta_1, \theta_2) < 1e-5$ | 1 | 2 | 3 | 4 | 5 | 6 | 7 | 8 | 10 | 12 | | 16 |

| $i$ | 13 | 14 | 15 | 16 | 17 | 18 | 19 | 20 | 21 | 22 | 23 | 24 |
|---|---|---|---|---|---|---|---|---|---|---|---|---|
| $j : \phi_{\text{MATCH}}^{(i,j)}(\theta_1, \theta_2) < 1e-5$ | 17 | 18 | 19 | 20 | 21 | 22 | 25 | 27 | 28 | 29 | 31 | 32 |

*Table 11.* $\theta_1 = $ `Sheared-LLaMa 1.3B` blocks matched with $\theta_2 = $ `Llama-2-7B` blocks using $\phi_{\text{MATCH}}$

Finally, we compare Llama 3.1 8B with `nvidia/Llama-3.1-Minitron-4B-Depth-Base`, a pruned model by reducing from 32 to 16 Transformer blocks and are able to identify the likely shared blocks.

| $i$ | 1 | 2 | 3 | 4 | 5 | 6 | 7 | 8 | 9 | 10 | 11 | 12 | 13 | 14 | 15 | 16 |
|---|---|---|---|---|---|---|---|---|---|---|---|---|---|---|---|---|
| $j : \phi_{\text{MATCH}}^{(i,j)}(\theta_1, \theta_2) < 1e-90$ | 1 | 2 | 3 | 4 | 5 | 6 | 7 | 8 | 9 | 10 | 11 | 12 | 13 | 14 | 15 | 32 |

*Table 12.* $\theta_1 = $ `nvidia/Llama-3.1-Minitron-4B-Depth-Base` blocks matched with $\theta_2 = $ `Llama-2-7B` blocks using $\phi_{\text{MATCH}}$

## F.4. MLP Distillation Experiments

As we mentioned in section 3.2.2, we present further results on experiments where we distill a model without a GLU MLP and then test the efficacy of our approach. These models do not use GLU MLPS (instead, a different feed-forward network) and a GLU MLP is distilled as the first FFN using Algorithm 5. The cases of two non-independent models still have very small values of $\phi_{\text{MATCH}}^{(1)}$.

| $\theta_1$ | $\theta_2$ | Independent? | $\phi_{\text{MATCH}}^{(1)}$ |
|:---:|:---:|:---:|:---|
| gpt2 | GPT2_PMC | ✗ | 3.034e-61 |
| gpt2 | artgpt2tox | ✗ | 1.049e-75 |
| gpt2 | distilgpt2 | ✗ | 1.079e-63 |
| Llama-3.2-1B | Llama-3.2-3B | ✗ | 2.011e-70 |
| openai-gpt | gpt2 | ✓ | 0.359 |
| openai-gpt | distilgpt2 | ✓ | 0.770 |
| gpt | Llama-3.2-1B | ✓ | 0.481 |

*Table 13.* $\phi_{\text{MATCH}}^{(1)}$ on models distilled with a GLU MLP.

| Parameter name | $\theta$ | $\text{Rot}(\theta) = \theta'$ |
|:---:|:---:|:---:|
| embedding | $E$ | $E R_{\text{emb}}$ |
| input layernorm | $\gamma_{\text{input}, i}$ | $\gamma'_{\text{input}, i}$ |
| attention query matrix | $W_{Q,i}$ | $R_i\, W_{Q,i}\, \text{diag}(\gamma_{\text{input}, i})\, R_{\text{emb}}\, \text{diag}(\frac{1}{\gamma'_{\text{input}, i}})$ |
| attention key matrix | $W_{K,i}$ | $R_i\, W_{K,i}\, \text{diag}(\gamma_{\text{input}, i})\, R_{\text{emb}}\, \text{diag}(\frac{1}{\gamma'_{\text{input}, i}})$ |
| attention value matrix | $W_{V,i}$ | $W_{V,i}\, \text{diag}(\gamma_{\text{input}, i})\, R_{\text{emb}}\, \text{diag}(\frac{1}{\gamma'_{\text{input}, i}})$ |
| attention output matrix | $W_{O,i}$ | $R_{\text{emb}}^T\, W_{O,i}$ |
| post-attention layernorm | $\gamma_{\text{post-attn}, i}$ | $\gamma'_{\text{post-attn}, i}$ |
| MLP gate projection | $G_i$ | $G_i\, \text{diag}(\gamma_{\text{post-attn},i})\, R_{\text{emb}}\, \text{diag}(\frac{1}{\gamma'_{\text{post-attn}, i}})$ |
| MLP up projection | $U_i$ | $c_i U_i\, \text{diag}(\gamma_{\text{post-attn},i})\, R_{\text{emb}}\, \text{diag}(\frac{1}{\gamma'_{\text{post-attn}, i}})$ |
| MLP down projection | $D_i$ | $\frac{1}{c_i}\, R_{\text{emb}}^T\, D_i$ |
| final layernorm | $\gamma_{\text{final}}$ | $\gamma'_{\text{final}}$ |
| linear output | $O$ | $O\, \text{diag}(\gamma_{\text{final}})\, R_{\text{emb}}\, \text{diag}(\frac{1}{\gamma'_{\text{final}}})$ |

*Table 14.* Output-preserving rotation applied to a Llama-architecture model.

## G. Output-Preserving Transformations

An adversary could apply a particular rotation scheme by multiplying weight matrices by an orthogonal rotation matrix $U$ that will also preserve outputs. We describe such a transformation which breaks the invariants proposed by (Zeng et al., 2024) by manipulating layernorms. While this list may not be exhaustive, the following six transformations (with the first two described previously) "camouflage" the language model while preserving outputs:

T1. Permuting the rows of the embedding matrix (and subsequent matrices due to residual connections) by a permutation $\xi_{\text{emb}} \in \mathbb{R}^{d_{\text{emb}} \times d_{\text{emb}}}$

T2. Permuting the MLP matrices ($N$ different permutations for each Transformer block) by permutations $\xi_1, \ldots, \xi_N \in \mathbb{R}^{d_{\text{mlp}} \times d_{\text{mlp}}}$

T3. Rotating the embedding matrix (and subsequent matrices due to residual connections) by an orthogonal rotation matrix $R_{\text{emb}} \in \mathbb{R}^{d_{\text{emb}} \times d_{\text{emb}}}$

T4. Rotating the query and key attention matrices ($N$ different rotations for each Transformer block) by orthogonal rotation matrices $R_1, \ldots, R_N \in \mathbb{R}^{d_{\text{emb}} \times d_{\text{emb}}}$

T5. Replacing all layernorms (input, post-attention, final) with vectors in $\mathbb{R}^{1 \times d_{\text{emb}}}$ with non-zero elements

T6. Scaling the MLP matrices by a constant non-zero factor

Consider a model $\theta$ of Llama architecture (Appendix C). Consider orthogonal matrices $R_{\text{emb}}, R_1, \ldots R_{32}$ as described, as well as new layernorms $\gamma'_{\text{input},1}, \ldots, \gamma'_{\text{input},32}, \gamma'_{\text{post-attn},1}, \ldots, \gamma'_{\text{post-attn},32}$ in $\mathbb{R}^{1 \times d_{\text{emb}}}$ with non-zero elements. Finally, consider non-zero constants $c_1, \ldots, c_{32}$, which we use to transform the layernorms. We apply the rotation with these parameters to $\theta$, to get a new "rotated" model, $\text{Rot}(\theta)$. We generalize the set of transformations above as applying $\text{Rot}(\theta)$ to a model $\theta$".

We transform all the original matrices of $\theta$ as in Table 14 (for $i = 1, \ldots, 32$). Note that the transformations T1 and T2 are elements of $\Pi_{\text{emb}}$ and $\Pi_{\text{mlp}}$ and the remaining transformations T3 to T6 are described in Table 14. Importantly, T5 is the transformation that (Zeng et al., 2024)'s invariants are not robust to; our unconstrained setting test $\phi_{\text{MATCH}}$ is robust to all 6 transformations, which we show in Table 15.

## G.1. Breaking HuREF Invariants

Only transformations T3 and T5 are required to break the invariants from (Zeng et al., 2024). Their first invariant is $M_a = E(W_{Q,i})^T W_{K,i})E^T$ at layer $i$, and for $M'$ with an embedding matrix rotation $R_{\text{emb}}$ where the layernorms $\gamma_{\text{input},i}$ are replaced with $\gamma'_{\text{input},i}$, we have the invariant is

$$M_a = E'(W'_{Q,i})^T((W'_{K,i})^T)^T E'^T$$

$$M'_a = (ER_{\text{emb}})\left(\text{diag}(\frac{1}{\gamma'_{\text{input},i}})R_{\text{emb}}^T\text{diag}(\gamma_{\text{input},i})W_{Q,i}^T R_i^T\right)\left(R_i W_{K,i}\text{diag}(\gamma_{\text{input},i})R_{\text{emb}}\text{diag}(\frac{1}{\gamma'_{\text{input},i}})\right)(R_{\text{emb}}^T E)$$

$$= ER_{\text{emb}}\text{diag}(\frac{1}{\gamma'_{\text{input},i}})R_{\text{emb}}^T\text{diag}(\gamma_{\text{input},i})W_{Q,i}^T W_{K,i}\text{diag}(\gamma_{\text{input},i})R_{\text{emb}}\text{diag}(\frac{1}{\gamma'_{\text{input},i}})R_{\text{emb}}^T E,$$

and in general $M_a \neq M'_a$ unless the layernorm weights are equal constants. The other two invariants also do not hold due to changing the layernorms. (Note that our notation for Transformers is different than theirs.) Assuming in their invariant $M_f$ that $W_1$ and $W_2$ are the gate and down projection matrices of an MLP (this is not stated explicitly in the paper but can be inferred from experiments), the remaining invariants do not hold either.

Empirically, we compute the invariants between Llama2-7b and independently trained models and between Llama2-7b and rotated finetuned models (including Llama2-7b) in Table 15. We can see there is little distinction between the independent vs. non-independent model pairs.

| $\theta_1 = $ Llama-2-7b-hf, $\theta_2 = $ | Independent? | $M_a$ | $M_b$ | $M_c$ | $\phi_{\text{MATCH}}$ | $\phi_{U(\ell)}$ | $\phi_{H(\ell)}$ | $\phi_{\text{JSD}}$ |
|---|---|---|---|---|---|---|---|---|
| vicuna-7b-v1.5 | ✗ | 1.0 | 0.9883 | 0.9922 | $< \varepsilon$ | $< \varepsilon$ | $< \varepsilon$ | -10.874 |
| Nous-Hermes-llama-2-7b | ✗ | 1.0 | 1.0 | 1.0 | $< \varepsilon$ | $< \varepsilon$ | $< \varepsilon$ | -12.101 |
| llama-7b-hf | ✓ | 0.0884 | 0.0250 | 0.0400 | 0.049 | 0.595 | 0.253 | -11.102 |
| AmberChat | ✓ | 0.1289 | -0.0093 | 0.0198 | 0.941 | 0.460 | 0.279 | -10.281 |
| Openllama-v1 | ✓ | 0.1084 | 0.0076 | 0.0057 | 0.286 | 0.357 | 0.703 | -8.381 |
| Rotated Llama-2-7b-hf | ✗ | 0.0767 | 0.0908 | 0.1011 | $< \varepsilon$ | 0.517 | 0.323 | $-\infty$ |
| Rotated vicuna-7b-v1.5 | ✗ | 0.1553 | 0.0933 | 0.0977 | $< \varepsilon$ | 0.688 | 0.857 | -10.874 |
| Rotated Nous-Hermes-llama-2-7b | ✗ | 0.0332 | 0.0718 | 0.1060 | $< \varepsilon$ | 0.772 | 0.240 | -12.101 |

*Table 15.* Results for the three invariants $M_a, M_b, M_c$ from (Zeng et al., 2024) between Llama-2-7b-hf and independent and non-independent models.

## G.2. Invariance of Outputs under Rotation

These transformations are particularly important because they preserve outputs as we show in Theorem **??**, and hence generally can go undetected, though $\phi_{\text{MATCH}}$ is robust to them.

**Theorem 5.** *For any input sequence $X \in \{0,1\}^{n \times V}$, the outputs of models $\theta$ and $Rot(\theta) = \theta'$ are aligned, i.e. $f_{\text{LM}}(X; \theta) = f_{\text{LM}}(X; \theta')$.*

*Proof.* First, note that an element-wise product of two one-dimensional vectors is equivalent to multiplying by the diagonal matrix of the second vector, i.e. for $v, \gamma \in R^{1 \times m}$,

$$v * \gamma = v\text{diag}(\gamma).$$

We use this in our layernorm calculations.

Let the output from the unrotated embedding layer be $y = f_{\text{in}}(X, E) = EX$ (for $X \in \{0,1\}^{n \times V}$). Then the output from the rotated embedding layer is $y' = f_{\text{in}}(X, E') = (ER_{\text{emb}})(x) = yR_{\text{emb}}$. Now consider Transformer block $i$ with input $y$ and the rotated Transformer block with input $yR_{\text{emb}}$. $y$ is passed into the input layernorm, which returns

$$z = LN_i(y) = \frac{y}{\sqrt{\text{Var}(y) + \varepsilon}} \odot \gamma_{\text{input},i} = \frac{y}{\sqrt{\text{Var}(y) + \varepsilon}}\text{diag}(\gamma_{\text{input},i}).$$

The rotated input layernorm on $y'$ returns

$$z' = LN'_i(y') = \frac{y'}{\sqrt{\text{Var}(y') + \varepsilon}} \odot \gamma'_{\text{input},i} = \frac{yR_{\text{emb}}}{\sqrt{\text{Var}(yR_{\text{emb}}) + \varepsilon}} \odot \gamma'_{\text{input},i}$$

$$= \frac{y}{\sqrt{\text{Var}(y) + \varepsilon}}R_{\text{emb}}\text{diag}(\gamma'_{\text{input},i}) = z\,\text{diag}(\frac{1}{\gamma_{\text{input},i}})R_{\text{emb}}\text{diag}(\gamma'_{\text{input},i}),$$

which follows from $R_{\text{emb}}$ being orthogonal. Then we have the output from the unrotated self-attention is

$$w = \text{softmax}\left(\frac{zW_{Q,i}^T(zW_{K,i}^T)^T}{\sqrt{d_{\text{key}}}}\right)zW_{V,i}^TW_{O,i},$$

and the output from the rotated self-attention with input $z'$ is

$$\text{softmax}\left(\frac{z'(R_iW_{Q,i}\text{diag}(\gamma_{\text{input},\,i})R_{\text{emb}}\text{diag}(\frac{1}{\gamma'_{\text{input},\,i}}))^T(z'(R_iW_{K,i}\text{diag}(\gamma_{\text{input},\,i})R_{\text{emb}}\text{diag}(\frac{1}{\gamma'_{\text{input},\,i}}))^T)^T}{\sqrt{d_{\text{key}}}}\right)$$

$$z'(W_{V,i}\text{diag}(\gamma_{\text{input},\,i})R_{\text{emb}}\text{diag}(\frac{1}{\gamma'_{\text{input},\,i}}))^T(R_{\text{emb}}^TW_{O,i})^T$$

$$= \text{softmax}\left(\frac{z'\text{diag}(\frac{1}{\gamma'_{\text{input},\,i}})R_{\text{emb}}^T\text{diag}(\gamma_{\text{input},\,i})W_{Q,i}^TR_i^T(z'\text{diag}(\frac{1}{\gamma'_{\text{input},\,i}})R_{\text{emb}}^T\text{diag}(\gamma_{\text{input},\,i})W_{K,i}^TR_i^T)^T}{\sqrt{d_{\text{key}}}}\right)$$

$$z'\text{diag}(\frac{1}{\gamma'_{\text{input},\,i}})R_{\text{emb}}^T\text{diag}(\gamma_{\text{input},\,i})W_{V,i}^TW_{O,i}R_{\text{emb}}$$

$$= \text{softmax}\left(\frac{z'\text{diag}(\frac{1}{\gamma'_{\text{input},\,i}})R_{\text{emb}}^T\text{diag}(\gamma_{\text{input},\,i})W_{Q,i}^TW_{K,i}\text{diag}(\gamma_{\text{input},\,i})R_{\text{emb}}\text{diag}(\frac{1}{\gamma'_{\text{input},\,i}})(z')^T}{\sqrt{d_{\text{key}}}}\right)zW_{V,i}^TW_{O,i}R_{\text{emb}}$$

$$= \text{softmax}\left(\frac{zW_{Q,i}W_{K,i}^Tz^T}{\sqrt{d_{\text{key}}}}\right)zW_{V,i}^TW_{O,i}R_{\text{emb}}$$

$$= wR_{\text{emb}} = w'.$$

Then $y$ and $y'$ respectively from before the layernorm are added as residual connections as $v = y + w$ and $v' = y' + w' = vR_{\text{emb}}$. $v$ is passed into the post-attention layernorm, which returns

$$u = LN_i(v) = \frac{v}{\sqrt{\text{Var}(v) + \varepsilon}} \odot \gamma_{\text{post-attn},\,i} = \frac{v}{\sqrt{\text{Var}(v) + \varepsilon}}\text{diag}(\gamma_{\text{post-attn},\,i}).$$

Similar to the input layernorm, the rotated post-attention layernorm on $v'$ returns

$$u' = LN'_i(v') = \frac{v'}{\sqrt{\text{Var}(v') + \varepsilon}} \odot \gamma'_{\text{post-attn},\,i} = \frac{vR_{\text{emb}}}{\sqrt{\text{Var}(vR_{\text{emb}}) + \varepsilon}} \odot \gamma'_{\text{post-attn},\,i}$$

$$= \frac{v}{\sqrt{\text{Var}(v) + \varepsilon}}R_{\text{emb}}\text{diag}(\gamma'_{\text{post-attn},\,i}) = u\,\text{diag}(\frac{1}{\gamma_{\text{post-attn},\,i}})R_{\text{emb}}\text{diag}(\gamma'_{\text{post-attn},\,i}).$$

Then the output from the unrotated MLP layer on $u$ is

$$t = [\sigma(uG_i^T) \odot (uU_i^T)]D_i^T$$

and the output from the rotated MLP on $u'$ is

$$t' = [\sigma(u'(G_i\text{diag}(\gamma_{\text{post-attn},i})R_{\text{emb}}\text{diag}(\frac{1}{\gamma'_{\text{post-attn},i}}))^T \odot (u'(c_iU_i\text{diag}(\gamma_{\text{post-attn},i})R_{\text{emb}}\text{diag}(\frac{1}{\gamma'_{\text{post-attn},i}}))^T)](\frac{1}{c_i}R_{\text{emb}}^TD_i)^T$$

$$= [\sigma(u\,\text{diag}(\frac{1}{\gamma_{\text{post-attn},i}})R_{\text{emb}}\text{diag}(\gamma'_{\text{post-attn},i})\text{diag}(\frac{1}{\gamma'_{\text{post-attn},i}})R_{\text{emb}}^T\text{diag}(\gamma_{\text{post-attn},i})G_i^T) \odot$$

$$(c_iu\,\text{diag}(\frac{1}{\gamma_{\text{post-attn},i}})R_{\text{emb}}\text{diag}(\gamma'_{\text{post-attn},i})\text{diag}(\frac{1}{\gamma'_{\text{post-attn},i}})R_{\text{emb}}^T\text{diag}(\gamma_{\text{post-attn},i}))U_i^T]\frac{1}{c_i}D_i^TR_{\text{emb}}$$

$$= [c_i\sigma(uG_i^T) \odot (uU_i^T)]\frac{1}{c_i}D_i^TR_{\text{emb}} = tR_{\text{emb}}.$$

Then the output from the self-attention is added as a residual connection, and the final output from the unrotated Transformer block is $s = t + v$, and the output from the rotated Transformer block is $s' = t' + v' = sR_{\text{emb}}$.

Suppose $a$ is the output after all Transformer layers in $\theta$ and $a'$ is the output after all Transformer layers in $\theta'$. Then the outputs after the final layernorms are

$$b = \frac{v}{\sqrt{\text{Var}(a) + \varepsilon}}\text{diag}(\gamma_{\text{final}})$$

$$b' = b \operatorname{diag}(\frac{1}{\gamma_{\text{final}}}) R_{\text{emb}} \operatorname{diag}(\gamma'_{\text{final}}),$$

and the logits from the linear output layer are

$$bO^T = b \operatorname{diag}(\frac{1}{\gamma_{\text{final}}}) R_{\text{emb}} \operatorname{diag}(\gamma'_{\text{final}}) \operatorname{diag}(\gamma_{\text{final}}) R_{\text{emb}}^T \operatorname{diag}(\frac{1}{\gamma'_{\text{final}}}) O^T$$
$$= b'(O')^T,$$

which are the same for both models. $\qquad\square$

We attempted to undo such a transformation that an adversary may apply by solving the least squares problem: We solve for a rotation $A$ that minimizes $|AX - Y|$ where $X$ is a weight matrix of the first model and $Y$ is the corresponding weight matrix of the second model. Although this will provide a potential rotation to undo this transformation, we find that this solution will also find a matrix $A$ that aligns two independent model pairs as well. This makes undo-ing the rotation this way unreliable. The same holds for $X$ and $Y$ that are activations over multiple inputs.

