# OpenReview forum: "Independence Tests for Language Models"
_ICML.cc/2025/Conference — ICML 2025 spotlightposter_

### Official Review · Reviewer_wPPE · 2025-02-23

**Overall Recommendation:** 3

**Summary:**

This paper introduces a method to assess whether two large language models (LLMs) are independent or if their training procedures exhibit dependencies. The core idea is based on the principle that if two LLM weights are independent, the distribution of differences between arbitrary permutations of their weights should be uniform. Conversely, if the models are dependent, the differences between their original weights will be significantly smaller than those between permuted versions of their weights. Motivated by this observation, the authors propose a method to compute p-values from the distribution of weight differences, which quantify the probability of the two models being independent. The paper presents both the conceptual framework and algorithmic implementation of their approach. Empirical results on some cases demonstrates its effectiveness in detecting dependencies between LLMs.

**Claims And Evidence:**

The submission makes claims in Theorem 3 that require further clarification and evidence. Specifically:
- The theorem does not guarantee that the parameters (thetas) are independent when two matches are independent, which raises concerns about the potential for a high Type-II error in the proposed test. This limitation is not sufficiently addressed, and the evidence supporting the test's robustness in such scenarios is unclear.
- The equivariant-type condition, which is central to the theorem, may not hold in practice. The submission does not provide adequate justification or empirical validation for this condition, casting doubt on the generality and applicability of the claimed results.

**Essential References Not Discussed:**

This paper employs a permutation-based technique to test independence. However, recent relevant works, such as [1-2], are omitted from the discussion.

Reference

- [1] Berrett, Thomas B., Ioannis Kontoyiannis, and Richard J. Samworth. "Optimal rates for independence testing via U-statistic permutation tests." The Annals of Statistics 49.5 (2021): 2457-2490.

- [2] Kim, Ilmun, Sivaraman Balakrishnan, and Larry Wasserman. "Minimax optimality of permutation tests." The Annals of Statistics 50.1 (2022): 225-251.

**Experimental Designs Or Analyses:**

I have concerns about the soundness and validity of the experimental design and analyses in the paper:

- Given that the proposed method is agnostic to neural networks, it would be beneficial to include experiments with simple neural network architectures to study Type-I and Type-II errors. This would help validate the method's effectiveness and robustness in a controlled setting, which is currently missing from the experimental design.

- $T$ used in the experiments appears to be too small, which may limit the reliability of the results.

**Methods And Evaluation Criteria:**

I have some concerns about the proposed methods and evaluation criteria for the problem at hand.

First, the definition of independence in Section 2.1, presented in Equation 1, is not well-defined. The notation $\theta_1 \perp \theta_2$ is used without a clear explanation of it. For instance, does it imply statistical independence, zero mutual information, or some other form of independence? This lack of clarity makes it difficult to assess the validity of the proposed test. Furthermore, the authors suggest that if (non-independent initializations), then (non-independent final weights). However, they also imply that if or even $A_1 = A_2$ (i.e., non-independent or identical training procedures), then $A_1(\theta_1^0) = A_2(\theta_2^0)$ (i.e., independent final weights), without providing sufficient justification.

Second, the evaluation criteria do not seem adequate for the problem. For example, the authors do not report results for Type-I error rates or statistical power, which are critical for assessing the reliability and effectiveness of the proposed test. Including these metrics would provide a more comprehensive evaluation of the method's performance.

**Other Comments Or Suggestions:**

Typos:

- Section 3.1, the first sentence: "We first validate validate the effectiveness"

**Other Strengths And Weaknesses:**

- Strengths: The paper is mathematically thorough and bases its thesis in well formulated assumptions. The idea is interesting and the method relatively simple and inexpensive, which may be attractive for implementation.

- Weaknesses:
  - The significance of independence testing for language models is not sufficiently motivated or clarified. The paper would benefit from a more detailed and board discussion of why this problem is important in the context of language models and their applications.
  - The clarity of writing could be improved. The presentation of ideas is often unclear, which hinders the reader's ability to fully grasp it.

**Questions For Authors:**

Please refer to **Claims and Evidence**, **Methods and Evaluation Criteria**, and **Weaknesses**.

**Relation To Broader Scientific Literature:**

This paper's contributions relate to the broader literature on independence testing, particularly for large language models (LLMs). It may be helpful for applications like LLM-based ensemble methods and model voting.

**Theoretical Claims:**

Yes, I have reviewed the proofs provided in Appendix B of the paper. I have the following concerns regarding their correctness.

1. The definition of independence in Appendix B is not well-defined. The lack of a precise and rigorous formulation makes it difficult to assess the validity of the theoretical claims and proofs that rely on this definition.

2.  The proof appears to be overly general and does not leverage any specific characteristics of language models. While this generality might seem advantageous, it raises questions about whether the proof is sufficiently tailored to the problem at hand. The absence of model-specific considerations limits the depth of the theoretical insights and their relevance to the application domain.

---

> ### Author Rebuttal · Authors · 2025-03-31
>
> We thank the reviewer for their time and address some of their concerns below. We will add the references suggested.
>
> $\textbf{Claims and Evidence Comment 1}$
>
> For the constrained test, we in fact guarantee that the test leads to exact p-values under the null hypothesis when two models are independent, i.e. two independent models will not lead to a low p-value.
>
> But with regards to a Type II error, we agree that our tests do not inherently guarantee that two non-independent models will always lead to a low p-value. However, empirically we find this is true in all our experiments. Specifically, Figures 5 and 6 in Appendix E show that on all (69) dependent Llama 7B pairs, the tests $\phi_{U}$ and $\phi_H$ yield p-values less than 2.2e-308 (which is our Type-II error rate, i.e. the maximum p-value we observe in the case that the null hypothesis (independence) does not hold). The same holds for $\phi_\text{MATCH}$ (Figure 7). We believe that this is sufficiently addressed through evidence from the Llama 7B and 70B results. We note that with regards to a Type I error, that our theorem does guarantee a uniform distribution for two independent models under the equivariance assumption, which we discuss more below.
>
> $\textbf{Claims and Evidence Comment 2}$
>
> Standard machine learning algorithms such as SGD, which are the most common algorithms for training language models follow the equivariance condition, as the gradients are permutation-equivariant, explained briefly in Example 2. We also emphasize that only one of the models needs to satisfy these assumptions, so a trusted model’s developer (who used SGD for example) can run this test without assumptions on the training strategy for another model.
>
> $\textbf{Methods and Evaluations Criteria}$
>
> The independence is statistical independence of two random variables, $\theta_1$ and $\theta_2$ (equivalently, zero mutual information), i.e. $X \perp Y$ if and only if $P(X|Y) = P(X)$. We will clarify this in a footnote in updated writing.
>
> In this case, if $\theta_1^0$ and $\theta_2^0$ are independent random variables, then by the post-processing inequality, since $A$ is a (deterministic) function, then $A(\theta_1^0)$ and $A(\theta_2^0)$ are also independent. We will add these details and this explanation.
>
> The Type-I error rate is the $\alpha$ threshold determined by the test user, since our tests give p-values. If the user chooses a threshold of $\alpha = 0.0001$ for example, then the Type-I error rate is 0.0001. This result is a consequence of our Theorem 1 with the equivariance condition. Further, we also plotted the null distribution for the 141 independent model pairs and found that the values are uniformly distributed, with exact values shown in Figure 6, for example.
>
> In our experiments (Figures 5, 6, 7 in the Appendix) (using ground truth from Hugging Face), choosing $\alpha =$ 1e-307 would yield a Type-I error rate of 1e-307 and a Type-II error rate of 0.
>
> $\textbf{Experimental Designs Or Analyses}$
>
> Our experiments are already conducted on neural networks; $\phi_U$, $\phi_H$, and $\phi_\text{MATCH}$ are all defined over the individual MLP component of the Transformer models, which are in principle simple neural networks. For example, $\phi_U$ uses the weights of the up-projection matrix, i.e. one layer of a neural network.
>
> By varying model size and the dimensions of the weight / activation matrices from 1B to 7B to 70B models, we test the effectiveness and robustness of the test; we also discuss Type-I and Type-II errors above.
>
> Our tests $\phi_U$, $\phi_H$, and $\phi_\text{MATCH}$ do not use a $T$ value, as they follow the general form provided in Equation (2). In principle, $T$ is actually the total number of permutations, which is why we get a very low p-value like e-308.
>
> $\textbf{Theoretical Claims comment and Weakness 1}$
>
> We agree that our proof holds for a broad class of machine learning models and neural networks. We intentionally chose this to highlight the generality of our method, but we focused our empirical experiments on language models due to their widespread re-use and the subsequent intellectual property concerns. Specifically, language model capabilities and pretraining costs are growing, which makes models more at risk of being stolen. Furthermore, many parties will fine tune an open-source model for a downstream task rather than pretraining their own model. We briefly discussed this in the second paragraph of the introduction but can further expand on the motivation. But, we note that the unconstrained test is geared towards language models (GLU MLPs).
>
> We acknowledge your concern about model-specific considerations and would appreciate it if you could point out any aspects that would significantly benefit from incorporating more language model specific characteristics, and if there was a specific issue you found with the proof.
>
> We will also work on the clarity of writing and appreciate the in-depth feedback.
>
> Thank you!

---

> > ### Comment · Reviewer_wPPE · 2025-04-04
> >
> > The authors confused several statistical concepts. The p-values (random variables) are not equivalent to Type-I and Type-II error (fixed constants). And p-values themselves cannot be used to estimate the Type-I and Type-II errors.
> >
> > If independence is defined as zero mutual information, the proposed method may be problematic. The setup involves two model parameters theta1 and theta2, each trained only once, yielding a single realization per parameter. Consider a simplest case, theta1 and theta2 jointly comes from a bivariate gaussian distribution, and we have just observation from the Gaussian distribution. With only one observation, Type-II error can be uncontrolled large even for parametric test. The absence of empirical results on Type-I/II error rates exacerbates these concerns.
> >
> > The proof of the theorem does not clearly leverage any specific features of language models that might offer sharper results (or weaker assumption). The result itself seems counterintuitive: the pseudo-observations used for testing are not independent, yet the theorem does not address how dependence affects Type-I error control.

---

> > > ### Author Response · Authors · 2025-04-09
> > >
> > > We thank the reviewer for their additional time, and address comments from the new rebuttal response. We hope this clarifies some of the discussion points.
> > >
> > > 1. "The authors confused several statistical concepts. The p-values (random variables) are not equivalent to Type-I and Type-II error (fixed constants). And p-values themselves cannot be used to estimate the Type-I and Type-II errors."
> > >
> > > We do not equate p-values with type-I/II errors. Rather, because we guarantee that Algorithm 1 test yields a valid p-value (i.e., the output of Algorithm 1 will be uniformly distributed between 0 and 1 under the null hypothesis), if we define a test based on thresholding the output of the algorithm by some value $\alpha \in [0,1]$ (i.e., if the output is larger than $\alpha$ then the test decides the two models are independent) then the type I error of this test will be $\textbf{exactly}$ $\alpha$. Because we have this guarantee for type I error, we focus our empirical evaluations in the constrained setting on type II error. In the unconstrained setting, we evaluate both errors since we no longer no longer have guaranteed control over either.
> > >
> > > 2. "If independence is defined as zero mutual information, the proposed method may be problematic."
> > >
> > > Independence of two random variables $\textbf{is equivalent}$ to those two random variables having zero mutual information (i.e., two random variables are independent if and only if they have zero mutual information. This follows directly from the definition of mutual information and the strict convexity of the map $x \mapsto -\log x$ (strict convexity implies the KL divergence between two random variables is 0 if and only if they are equal in distribution).
> > >
> > > 3. "The setup involves two model parameters theta1 and theta2, each trained only once, yielding a single realization per parameter."
> > >
> > > The goal of statistical inference is to draw conclusions about a random variable given a realization of the random variable (i.e., a sample). We adopt this familiar goal in our work.
> > >
> > > 4. "Consider a simplest case, theta1 and theta2 jointly comes from a bivariate gaussian distribution, and we have just observation from the Gaussian distribution. With only one observation, Type-II error can be uncontrolled large even for parametric test."
> > >
> > > We are not sure what task is being referenced here and the relevance to our work, but we would be happy to discuss further upon clarification. We certainly agree there are many tasks that are not achievable from one observation (e.g., two observations are required to obtain an unbiased estimator for the variance of a distribution).
> > >
> > > 5. "The absence of empirical results on Type-I/II error rates exacerbates these concerns."
> > >
> > > We empirically evaluate both type I/II errors in the unconstrained setting, and we evaluate type II errors in the unconstrained setting (see above for an explanation of why it would be redundant to evaluate type I errors in the constrained setting).
> > >
> > > 6. "The proof of the theorem does not clearly leverage any specific features of language models that might offer sharper results (or weaker assumption)."
> > >
> > > The result of the theorem cannot be any sharper: we prove Algorithm 1 yields an $\textbf{exact}$ p-value, so there is no bound to improve. We agree the proof does not leverage specific features of language models. This property is a strength rather than a weakness of the theorem: leveraging specific features of language models would necessarily require $\textbf{stronger assumptions}$ (for starters, we would need to assume $\theta_1$ and $\theta_2$ are language models).
> > >
> > > 7. "the pseudo-observations used for testing are not independent, yet the theorem does not address how dependence affects Type-I error control."
> > >
> > > As we discuss in the main body (e.g., Abstract, Introduction, and Section 2.2.1) and proof of the theorem (Appendix A), we crucially use the fact that the permuted models are $\textbf{exchangeable}$ with the original model (despite not being independent) to prove the theorem. See [1] for a definition of exchangeability.
> > >
> > > [1] https://en.wikipedia.org/wiki/Exchangeable_random_variables

---

### Official Review · Reviewer_RDaz · 2025-03-13

**Overall Recommendation:** 4

**Summary:**

The paper investigates a method to determine whether two models’ weights were trained independently (i.e., from different random initializations) or if one model was derived from the other through fine-tuning, pruning, or partial reuse. This is framed as a hypothesis test for independence between two sets of model weights.

The study considers two settings:
- Constrained setting: Both models have the same architecture. The authors assume the training process is equivariant to permutations of the hidden units, allowing them to compute exact p-values under the null hypothesis of independent training. They validate this method on 21 open-weight models and correctly identify all non-independent pairs.
- Unconstrained setting: Models can have different architectures. The authors develop a robust test based on aligning hidden activations, which remains effective despite architectural changes or adversarial modifications. Though this test does not produce exact p-values, it empirically behaves like one and can even pinpoint specific model components that are shared or derived.

Overall, the authors claim that the proposed methods reliably distinguish independent models from non-independent ones, even in cases where dependencies are obscured by architectural modifications or selective weight reuse.

**Claims And Evidence:**

The paper provides strong empirical evidence for the effectiveness of its proposed methods, particularly in detecting non-independent model pairs and identifying shared components.

**Essential References Not Discussed:**

None that I know of.

**Experimental Designs Or Analyses:**

Overall, the experiments conducted by the author is sound, tests were conducted across a wide range of models (mostly those related to Llama). For the unconstrained case, it would have been interesting to also explore other model families such as the Microsoft's Phi family.

**Methods And Evaluation Criteria:**

Yes.

**Other Comments Or Suggestions:**

None.

**Other Strengths And Weaknesses:**

None.

**Questions For Authors:**

None.

**Relation To Broader Scientific Literature:**

The paper contributes to the broader literature on intellectual property protection and model fingerprinting by demonstrating that model weights themselves can serve as a fingerprint for tracing model lineage. Unlike prior work that relies on embedding traceable signals in model outputs or specific responses, this study shows that statistical tests on model weights can effectively determine whether a model was trained independently or derived from another. This insight enhances provenance tracking and provides a new tool for enforcing licensing restrictions and protecting intellectual property in machine learning.

**Theoretical Claims:**

No issues as far as I know.

---

> ### Author Rebuttal · Authors · 2025-03-31
>
> We thank the reviewer for their time and positive feedback!
>
> We find the tests also work on smaller-scale models such as the Phi-3 family. Both $\phi_\text{CSH}$ and $\phi_\text{MATCH}$ return a statistic approximately 1e-308 (aggregated with Fisher’s method) on the fine-tuned model pair microsoft/Phi-3.5-mini-instruct and numind/NuExtract-v1.5 (3.8B parameters). We are also happy to add more experiments on other model families and will update our experiment section in our paper
>
> Thank you!

---

### Official Review · Reviewer_L4qp · 2025-03-15

**Overall Recommendation:** 3

**Summary:**

This paper introduces a rigorous statistical framework for testing whether two language models were trained independently. Concretely, the authors propose hypothesis tests in both constrained and unconstrained settings. The constrained setting assumes known model architecture and training conditions, allowing for exact p-value computation through simulations of exchangeable copies of each model. The unconstrained setting removes these assumptions, making the test robust to adversarial modifications that preserve model outputs but alter internal weight structures. The proposed methods are validated on many open-weight models.

##  update after rebuttal
 After reviewing the rebuttal addressed to me and those for other reviewers, I am willing to maintain my score.

**Claims And Evidence:**

Yes, the claims made in the submission appear to be supported by clear and convincing evidence.

**Essential References Not Discussed:**

No, the paper includes essential references.

**Experimental Designs Or Analyses:**

Yes. The experimental designs and analyses appear to be sound.

**Methods And Evaluation Criteria:**

Yes, the proposed method makes sense for the independence testing of two language models.

**Other Comments Or Suggestions:**

No

**Other Strengths And Weaknesses:**

Strengths
- This paper is well written and its structure is clear.
- The paper frames model independence as a hypothesis testing problem.
- In constrained setting: this paper uses exchangeable model copies under specific assumptions to compute exact p-values.

Weaknesses:

- The assumption for permutation-equivariant training may not always hold in real-world applications.
- While the unconstrained test is empirically robust, theoretical guarantees are lacking.
- More importantly, the assumption that the learning algorithms are deterministic functions is seriously inconsistent with the facts. Obviously, the outputs of the learning algorithms are primarily influenced by the training data and thus are random.

**Questions For Authors:**

No

**Relation To Broader Scientific Literature:**

This paper aims to address a fundamental question in model provenance and intellectual property protection.

**Theoretical Claims:**

Yes, I check some proofs, including Theorems 1, 2 and 3.

---

> ### Author Rebuttal · Authors · 2025-03-31
>
> We thank the reviewer for their time and positive feedback!
>
> $\textbf{Weakness 1}$
>
> Thank you for bringing up this concern. Standard machine learning algorithms such as SGD  which are the most common algorithms for training language models follow the equivariance condition, as the gradients are permutation-equivariant, explained briefly in Example 2. We also emphasize that only one of the models needs to satisfy these assumptions, so a trusted model’s developer (who used SGD for example) can run this test without assumptions on the training strategy for another model.
>
> $\textbf{Weakness 2}$
>
> Yes, we agree, and we do not claim our test is in fact robust to all adversarial attacks. We have at least empirically validated that our test is robust to a superset of attacks over prior work (Zeng et al. 2024).
>
> $\textbf{Weakness 3}$
>
> We agree that the final model weights are heavily influenced by the training data. However, in our definition of a learning algorithm (Section 2.1) we write “$A$ includes the choice of training data…” Given the training data, minibatch ordering, and other parameters, then in fact $A$ is deterministic and it is possible to fully reproduce the learning algorithm given the initial weights.
>
> This is an abstraction to simplify our framework and may be an unconventional way to describe learning algorithms, so we will add more clarification. We also acknowledge there are other sources of randomness which deterministic functions with a fixed seed do not capture such as dropout; thus, in Appendix A, we state and prove a more general version of Theorem 1 for randomized learning algorithms, for which our conclusions still hold.
>
> Thank you, and we are happy to answer more questions!

---

### Official Review · Reviewer_H5os · 2025-03-19

**Overall Recommendation:** 3

**Summary:**

The paper addresses the large model independence test: given the weights of two language models, can we determine if they were trained independently or if one model’s weights are derived from the other? Leveraging permutation invariance and equivariance in MLP neurons, it provides exact p-values. Extensive evaluations on open-weight models show that the test performs effectively.

**Claims And Evidence:**

Overall, the claims made in the paper are consistently supported. The strong statements are backed by theoretical proof or experiments. I did not find any major claim that is unsupported.

**Essential References Not Discussed:**

Some related works about permutation tests may help unfamiliar readers to better understand Algorithm 1, e.g., [1, 2].

References:
1. Phillip Good. Permutation Tests: A Practical Guide to Resampling Methods for Testing Hypotheses.
2. E.L. Lehmann, Joseph P. Romano. Testing Statistical Hypotheses.

**Experimental Designs Or Analyses:**

1. The evaluation primarily focuses on the Llama-7B architecture, which raises concerns about the generalizability of the proposed methods. Expanding the experiments to include a broader range of training regimes or adversarial scenarios would enhance confidence in their robustness.

2. While baselines such as $\phi_{JSD}$ and $\phi_{l_2}$ are included, the comparison to Zeng et al. (2024) is insufficiently addressed in the main text. A more thorough discussion of how the proposed methods outperform or complement prior work would strengthen the significance of the claims.

**Methods And Evaluation Criteria:**

1. The paper employs cosine similarity as a core metric for comparing model parameters. However, its reliance on linear relationships may overlook nonlinear dependencies between models. This limitation could lead to false negatives, particularly in the unconstrained setting where architectural differences and adversarial modifications are more prevalent, and the paper does not explore alternative metrics to address this gap.
2. Additionally, while the unconstrained setting handles size inconsistencies via zero-padding, this approach may introduce bias or reduce sensitivity when the dimensional disparity is significant. The effectiveness of zero-padding is demonstrated empirically (e.g., Llama-3.1-8B vs. Llama-3.2-3B), but the paper lacks analysis of its impact under extreme size mismatches or discussion of alternative alignment strategies, potentially compromising robustness in broader scenarios.
3. In the unconstrained setting, the paper employs a matching approach via the MATCH algorithm (Algorithm 2) to compare $\theta_1$ and $\theta_2$ despite potential dimensional inconsistencies, using zero-padding to align matrix sizes. While this method effectively identifies dependencies in experiments, its reliance on a strict one-to-one correspondence may be inadequate when the true relationship between models is one-to-many or many-to-many, such as in scenarios involving pruning, expansion, or complex retraining. Such relationships, which do not align with the bijective assumption of MATCH, could lead to false negatives, especially when dimensional disparities are significant, as zero-padding might obscure nuanced dependencies. The paper does not explore these possibilities or test the method’s robustness against non-bijective dependencies, limiting its applicability to more intricate model relationships.

**Other Comments Or Suggestions:**

1. Adding the mathematical definition of "exchangeable" is better.
2. The assumptions of $\Pi$-equivariance and $\Pi$-invariance are mathematically clear but may be difficult to readers unfamiliar with deep learning symmetries. More intuitive explanations or examples could improve accessibility.

**Other Strengths And Weaknesses:**

Strengths:
1. The problem of testing model independence has significant implications.
2. The methods are computationally efficient (e.g., avoiding full retraining by using permutations or proxy models), making them feasible for large-scale models.
3. The formal definitions (e.g., $\Pi$-invariance, $\Pi$-equivariance) and theorems provide a solid theoretical backbone.

Weaknesses:
1. From a practical application perspective, I think determining the direction of causal relationships between large models may be more meaningful than merely testing for the existence of a dependency. However, the latter remains a highly important and intriguing problem.
2. The running time of the test is also a very important practical consideration. Authors need to explicitly report needed times.
3. In the unconstrained setting, Algorithm 5’s proxy GLU MLP construction depends on the choice of hidden dimension $h$ and input distribution $P$. The paper offers little insight into their impact and how they were selected. The sensitivity of the method to these hyperparameters is also unclear.

**Questions For Authors:**

1. The motivation for choosing gate and up projections in $\phi_{MATCH}$ (Section 2.3.1) is described as a conjecture rather than a derived principle, which may leave readers questioning its robustness. Could you provide more justification?
2. In real-world applications, how to set significance level $\alpha$?
3. When the true relationship between sub-models is one-to-many or many-to-many, which do not align with the bijective assumption of MATCH, is there any better solution?

**Relation To Broader Scientific Literature:**

This contribution bridges classical statistical tools with modern deep learning to apply in a novel real-world problem, language model independence tests, which adapt permutation methods to a new domain while leveraging neural network symmetries.

**Theoretical Claims:**

I checked the proofs for Theorems 1-3 in the constrained setting. The logic is correct.

One issue is that the authors treat learning algorithms as deterministic functions, ignoring the possible randomness of dropouts, etc. This assumption limits the practical applicability of Theorem 1. Although partly discussed in Appendix A, the main text should provide a clear discussion of this limitation.

---

> ### Author Rebuttal · Authors · 2025-03-31
>
> We thank the reviewer for their time and feedback! We will add the references mentioned.
>
> $\textbf{Experimental Designs Or Analyses Comment 1}$
>
> We report experiments on Llama 70B, the hybrid StripedHyena and Mistral model, and (distilled) GPT 2 models in Table 5, 9, and 15  and find our statistics work for these different architectures as well. In Reviewer RDaz's comment, we also experiment with smaller Phi models and find the test holds high power for those models too. We believe these models likely encompass a broad range of training regimes.
>
> In Table 7 of the Appendix, we also run our tests on adversarially-transformed models and show how our unconstrained test is robust to the transformation (whereas prior work is not). If there are other model architectures or families that would be beneficial, we would be happy to add those experiments!
>
> $\textbf{Experimental Designs Or Analyses Comment 2}$
>
> We include more discussion and experiments in Appendix F.1 (HuREF Invariants) but are happy to move discussion to the main text. Specifically in Table 7, we demonstrate how our adversarial transformation can be used to break the HuREF invariants — each of the transformed Llama-2-7b-hf, vicuna-7b-v1.5, and Nous-Hermes-llama-2-7b models have low $M_a$, $M_b$, and $M_c$ values when compared with Llama-2-7b-hf, whereas our unconstrained statistic $\phi_\text{MATCH}$ gives a value of 2.2e-308.
>
> We also mention in the paragraph at Line 96 of the introduction that our tests yield p-values whereas Zeng et al.’s do not. Please let us know if there is more we can provide about comparison with Zeng et al. (2024).
>
> $\textbf{Weakness 1}$
>
> We agree. But in those cases, then causal relationships can be determined by first using our test, then using metadata, such as the dates of model releases.
>
> $\textbf{Weakness 2}$
>
> We report our times on a Nvidia RTX A6000. For $\phi_H$, the bottleneck is computing the forward pass to obtain the intermediate activations — and on two 7B models, the test on all 32 Transformer blocks combined takes on average less than 2 minutes. $\phi_\text{MATCH}$ requires the forward pass and aligning the activations; on dependent models, this will also take around 2 minutes total, whereas it may take 5-10 minutes per Transformer block for independent models. We believe this is reasonable computationally. We will include these details.
>
> $\textbf{Weakness 3}$
>
> The hidden dimension $h$ is determined by the model weights, i.e. for Llama 2-7B the hidden dimension is 11008. This dimension varies for the models we test (28672 for Llama 2-70B, 8192 for Llama 3.2-3B) but do not affect the strength of our tests.
>
> In our paper, we only report results for using WikiText as the input distribution, but in fact we are able to achieve similar performance using random tokens from the vocabulary as the input distribution. We will include further experiments and ablations on this distribution.
>
> $\textbf{Question 1}$
>
> We choose the gate and up projections of an MLP because they are two matrices that are combined via a direct product and activation function — which makes significant transformations to the weights difficult. However, the results in Figure 2 from the GLU MLP remain a conjecture. We reason that the gate and up projections are trained to be very aligned during training with a very large loss landscape for two independent models.
>
> $\textbf{Question 2}$
>
> Empirically, we find our results hold with a threshold of even 2.2e-308, and e-61 for the generalized unconstrained test where we distill the MLP. In practice, a third party or model provider may choose $\alpha =$ 1e-5 for example, or their own confidence level.
>
> $\textbf{Question 3}$
>
> We have tested non-bijective cases such as between Llama 3.1-8B and Llama 3.2-3B, where the dimension is reduced from 11008 to 8192 (more than 30%). We test $\phi_\text{MATCH}$ on many pruned models including the Nvidia Minitron models and Sheared Llama models in Appendix I.2, where the bijective assumption does not hold and find that the test is still strong.
>
> Also, if the reviewer is asking about many-to-many sub-models, we also run experiments, such as with the hybrid StripedHyena model, where only some layers are taken from the Mistral 7B model (i.e. embedding) but not others (MLP projections). We are happy to run more experiments or provide more clarification.
>
> We will also add more explanation with regards to equivariance and invariance. We appreciate the in-depth review and are happy to answer more questions!

---

### Official Review · Reviewer_5ZNo · 2025-03-25

**Overall Recommendation:** 4

**Summary:**

This paper proposes a statistical test for determining whether the initializations of two language models (really, "deep networks containing GLU MLPs", or even really slightly weaker than that) are independent or not, when treating the algorithms themselves (and any data they use, etc) as fixed. Exactly valid tests are developed under equivariance assumptions for the training algorithms; when this is not true, the paper proposes heuristic tests which seem to behave reasonably under the null.

**Claims And Evidence:**

Yes.

**Essential References Not Discussed:**

Theorem 1 is very closely related to e.g. Theorem 2 of [Hemerik and Goeman](https://arxiv.org/abs/1411.7565), who also cite earlier sources for roughly the same result. This is not a big deal, since that theorem in itself is simple and not a major contribution of the paper, but as that paper goes in more depth about related properties, it would be good to point readers to. (They assume a group structure on their transformations, implying invariance, while you directly assume invariance; I think this is the only difference.)

**Experimental Designs Or Analyses:**

The setups seem reasonable but I did not carefully examine the details.

**Methods And Evaluation Criteria:**

Yes.

**Other Comments Or Suggestions:**

- A few times the paper refers to "the set of permutations over the hidden units of the network"; this isn't really right (or what you do), since it wouldn't make sense to swap hidden units across layers/different modules.
- It would be good to add a sentence about the LAP algorithm of Ramshaw and Tarjan, even just saying that it is an algorithm for weighted matchings in bipartite graphs. You could easily save the space by not writing Algorithm 2 out in algorithm form and instead just put it in an equation display, since the algorithm form adds basically no information for this one.
- (extremely minor) "Our robust test reposes on the design" – this uncommon usage of "reposes" seems to be used mostly in theology and is probably unfamiliar to most ICLR readers."Relies on" would be far more typical.
- Your bib file is rather sloppy; you should e.g. remove most of the urls, especially the ones from Semantic Scholar.

**Other Strengths And Weaknesses:**

The proposed test is clever, relevant for an interesting problem, and appears to work in practice. I think the paper is worth publishing at ICML.

The proper interpretation of the null hypothesis, however, is subtle. The paper includes some discussion of these issues, and I don't think any of it is incorrect or even misleading, but thinking about independence where the data is fixed is somewhat unnatural, and I can easily see practitioners misinterpreting the outputs of the test.

As you point out, in the constrained setting you do have a valid test, and hence the only consideration is the power. You don't, though, give any formal discussion to the power of your tests. It seems like it might be possible to, for example, at least say something about how using Fisher's procedure across blocks relates in power to the permutation test? I'm not sure what "consistency" or similar properties would mean here since there's not iid data...but I believe there are probably some situations where despite the null hypothesis not holding, the test based on (2) has only trivial power. (I thought about it for a few minutes and couldn't come up with one in the constrained setting, but also couldn't convince myself that it's impossible; I think it likely is. In the unconstrained setting, doing so is trivial.)

> Due to the element-wise product operation, we conjecture that in general it is not possible to permute the rows of $G_i$ while preserving the output of $\theta_i$ without permuting the rows $U_i$ in the same way

"In general" is doing a lot of work here; it is easy to construct silly examples where this is not the case (e.g. take the activation $\sigma$ to always map to zero). This is not a big deal; for "reasonable models," this should be true. But it highlights the general issue in this paper that while some attempt is made at formality, there are many parts which are difficult to really formalize. I think that's probably inherent to the problem setting, but it does highlight how basically everything in the "unconstrained" setting is generally "reasonable" but does not have any strict definitions the way the "constrained" setting does (even if those definitions themselves require some thought to understand).

For the retraining and distilling test: it seems that this scheme could be tricked by first permuting the hidden units before and after the MLP, then retraining the MLP layer from scratch there, right?

**Questions For Authors:**

I don't think I have any specific questions, although I'd be interested to hear if you're able to say a little more about the power of the test in the constrained setting, or other points I raised above.

Update after rebuttal
------------------------

I accidentally posted the below comment in a way that you weren't able to see. I think it would be worth considering the question below for the next revision of your paper (whether camera-ready or a resubmit).

> Thanks for your reply; I remain happy with the paper.
>
> I realized in thinking slightly more about the setting today that: is there a particular reason you chose the statistic to compute the Spearman correlation between the best permutation and $[n]$? In particular, that induces a strong "locality" on any swaps: swapping hidden units 1 and 2 would "cost" much less to the statistic than swapping hidden units 1 and 1000. That doesn't seem to make sense to me, though, since there really isn't any locality in this sense to the network. In a related point, comparing whether the *best* match lines up is distinct from asking how much "better" the best match is than the identity. When permuting, I wonder whether it might not make more sense to ask about the ratio of the matching objective, or similar. This would probably be harder to avoid permuting with the approximate p-value from the Spearman correlation, though.

**Relation To Broader Scientific Literature:**

The proposed test differs from previous approaches for the same problem in a useful way, and seems to work better.

**Theoretical Claims:**

The proof techniques pass a "smell check" for me, but I did not carefully verify them.

---

> ### Author Rebuttal · Authors · 2025-03-31
>
> We thank the reviewer for their time and positive feedback! We will add the Hemerik and Goeman reference mentioned, thank you.
>
> About some of the weaknesses discussed:
>
> From our empirical results, the power of our constrained and unconstrained test is as low as 2.2e-308 — in the cases of rejecting the null hypothesis, the p-value derived is less than 2.2e-308. (See Figure 5, 6, 7 in the Appendix.) In particular, in the best case where $\theta_1 = \theta_2$, then the power will be exp(- # hidden units).
>
>
> The test could have trivial power in the case where the learning algorithms are constant functions, then the p-values will be non-significant for any initializations even if they are non-independent. However, this would never occur in practice for any non-trivially trained language model, and our tests have strong power for the wide array of language models we evaluate. We agree that the "in general" does a lot of work, so we only make our claims here via experimental results. We are happy to amend our writing to better calibrate to your feedback
>
> "For the retraining and distilling test": The unconstrained test is robust to permutations --- permuting the hidden units before and after the MLP (or if we permute the entire model) and distilling the permuted model would not change the efficacy of the test, as the gate and up projection matrices would need to share that original permutation.
>
> We will also apply the edits from "Other Comments or Suggestions." We are happy to answer more questions and appreciate the in-depth response!

---

### Decision · Program_Chairs · 2025-05-01

**Decision:**

Accept (spotlight poster)

**Comment:**

The paper tackles the question of whether two given models were trained independently (precisely, deep networks containing GLU MLPs). A statistical test with an exact p-value is developed under equivariance assumptions for the training algorithms. More precisely, the test leverages permutation invariance and equivariance in MLP units. A separate heuristic-based test was proposed if the assumptions are not met.

The paper addresses a timely problem. All the reviewers noted that this work proposes interesting approaches that are theoretically grounded. The contributed theoretical results (e.g., testing under equivariance, exact p-value computation) are solid (H5os, wPPE). The authors submitted a strong rebuttal that addresses most of the concerns raised, mostly about the permutation-equivariant assumptions, the assumption that the learning algorithms are deterministic, and other clarification questions  (e.g., runtime).

Much of the AC-reviewers discussion was on two topics: 1. potentially uncontrolled type-I error, 2. no guarantee that asymptotically type-II error goes to zero. These are valid concerns. The first concern was resolved by a consensus that the test (at least in the constrained setting) yields an exact p-value and so type-I error is guaranteed to be in control. The AC resolved the second concern by pointing out that the “asymptote” is not a well-defined concept here, unlike other more common testing scenarios where asymptote is tied to observing a larger sample. Here, sample means the two given models. Formally quantifying the type-II error will require the knowledge of the distribution under the alternative, which is non-trivial. The AC thinks that even without this analysis, as one of the few works that tackles this direction, the rest of the theoretical contributions are already sufficient.

AC recommendation: accept.


To the authors:

* Please include an impact statement in the final version. This is a must. Please see https://icml.cc/Conferences/2025/CallForPapers. This work is theoretical in nature. The AC does not expect this work to have direct ethical impacts or direct societal implications.